# T-cell calcium dynamics visualized in a ratiometric tdTomato-GCaMP6f transgenic reporter mouse

Tobias X Dong[1†], Shivashankar Othy[1†], Amit Jairaman[1], Jonathan Skupsky[1,2], Angel Zavala[1], Ian Parker[1,3], Joseph L Dynes[1], Michael D Cahalan[1,4]*

[1]Department of Physiology and Biophysics, University of California, Irvine, United States; [2]Department of Medicine, University of California, Irvine, United States; [3]Department of Neurobiology & Behavior, University of California, Irvine, United States; [4]Institute for Immunology, University of California, Irvine, United States

**Abstract** Calcium is an essential cellular messenger that regulates numerous functions in living organisms. Here, we describe development and characterization of 'Salsa6f', a fusion of GCaMP6f and tdTomato optimized for cell tracking while monitoring cytosolic $Ca^{2+}$, and a transgenic $Ca^{2+}$ reporter mouse with Salsa6f targeted to the Rosa26 locus for Cre-dependent expression in specific cell types. The development and function of T cells was unaffected in Cd4-Salsa6f mice. We describe $Ca^{2+}$ signals reported by Salsa6f during T cell receptor activation in naive T cells, helper Th17 T cells and regulatory T cells, and $Ca^{2+}$ signals mediated in T cells by an activator of mechanosensitive Piezo1 channels. Transgenic expression of Salsa6f enables ratiometric imaging of $Ca^{2+}$ signals in complex tissue environments found in vivo. Two-photon imaging of migrating T cells in the steady-state lymph node revealed both cell-wide and localized sub-cellular $Ca^{2+}$ transients ('sparkles') as cells migrate.
DOI: https://doi.org/10.7554/eLife.32417.001

*For correspondence:
mcahalan@uci.edu

†These authors contributed equally to this work

Competing interests: The authors declare that no competing interests exist.

## Introduction

Calcium ($Ca^{2+}$) is an essential second messenger responsible for a wide variety of cellular functions (*Berridge et al., 2000*; *Clapham, 2007*; *Berridge, 2012*). Through the use of synthetic small molecule $Ca^{2+}$ indicators such as fura-2 and fluo-4, imaging studies have greatly expanded our understanding of $Ca^{2+}$ signaling dynamics (*Tsien et al., 1982*; *Grynkiewicz et al., 1985*). However, such indicators cannot be targeted to specific subcellular compartments or cell populations, and are unsuitable for long-term studies due to leakage out of cells. Moreover, they often do not faithfully report pure cytosolic $Ca^{2+}$ signals, because of diffusion into cellular compartments such as the nucleus. One alternative to overcoming these limitations is with genetically encoded $Ca^{2+}$ indicators (GECIs), first developed two decades ago as FRET-based fluorescence probes (*Miyawaki et al., 1997*; *Romoser et al., 1997*; *Pérez Koldenkova and Nagai, 2013*). Key advantages to GECIs include the capability for genetic targeting to specific cell types or subcellular organelles, measuring local $Ca^{2+}$ levels by direct fusion to a protein of interest, modulation of expression levels by inclusion of an inducible promoter, and long-term studies due to continuous expression of the genetic indicator (*Miyawaki et al., 1997*; *Pérez Koldenkova and Nagai, 2013*). Despite these inherent advantages, the initial FRET-based GECI probes were not widely used as their performance fell far behind small molecule $Ca^{2+}$ indicators, particularly in $Ca^{2+}$ sensitivity, brightness, and dynamic range. Since then, successive rounds of design and contributions from multiple research groups have resulted in numerous variants of GECIs with high dynamic range and dramatically improved performance (*Baird et al., 1999*; *Nakai et al., 2001*; *Tian et al., 2009*; *Zhao et al., 2011*; *Akerboom et al.,*

**eLife digest** To help protect the body from disease, small immune cells called T lymphocytes move rapidly, searching for signs of infection. These signs are antigens – processed pieces of proteins from invading bacteria and viruses – which are displayed on the surface of so-called antigen-presenting cells. To visit as many different antigen-presenting cells as possible, T cells move quickly from one to the next in an apparently random manner. How T cells are programmed to move in this way is largely unknown.

The entry of calcium ions into cells, through channel proteins, triggers characteristic actions in many cells throughout the body. As such it is possible that the T cells' movements are related to calcium signals too. However, it was technically challenging to directly measure the amount of calcium in moving cells within the body.

To overcome this issue, Dong, Othy et al. genetically engineered mice to produce a new calcium-sensitive reporter protein in their T cells. The reporter, which was named Salsa6f, consisted of a red fluorescent protein fused to another protein that glows green when it binds to calcium ions. Measuring the ratio of red and green fluorescence gives a measure of the concentration of calcium ions inside the cell. In the absence of calcium signaling, the cells can still be tracked via the red fluorescence of Salsa6f. Importantly, the reporter did not affect the development or activity of the T cells in the mice. In a related study, Dong, Othy et al. then used their transgenic mice to ask whether calcium signals guide moving T cells as they search for antigens.

Future studies could use these transgenic mice to track the calcium ion concentration in numerous cell types. This would enable new approaches to relate the inner workings of cells to their behaviors in many different organ systems throughout the body.

DOI: https://doi.org/10.7554/eLife.32417.002

*2012*; *Akerboom et al., 2013*; *Chen et al., 2013*). Single fluorescent protein-based GECIs containing a circularly permutated green fluorescent protein (GFP) exhibit high brightness, fast response kinetics, and offer multiple color variants, including the GECO and the GCaMP series (*Tian et al., 2009*; *Zhao et al., 2011*; *Akerboom et al., 2012*; *Chen et al., 2013*). FRET-based GECIs have continued to evolve as well, with sequential improvements including incorporation of circularly permuted yellow fluorescent proteins (cpYFPs) to improve dynamic range in the yellow cameleon (YC) family (*Nagai et al., 2004*), use of troponin C as the $Ca^{2+}$ sensing element in the TN indicator family (*Heim and Griesbeck, 2004*), computational redesign of the calmodulin-M13 interface to increase the range of $Ca^{2+}$ sensitivity and reduce perturbation by native calmodulin in the DcpV family (*Palmer et al., 2006*), and complete redesign of the troponin C domain to increase response kinetics and reduce buffering of cytosolic $Ca^{2+}$ in the TN-XXL family (*Mank et al., 2006*; *Mank et al., 2008*).

The latest generation of GECIs have crossed key performance thresholds previously set by small-molecule indicators, enabling GECIs to be widely applied in diverse $Ca^{2+}$ imaging studies without sacrificing performance. Members of the GCaMP6 family are capable of tracking cytosolic $Ca^{2+}$ changes from single neuronal action potentials, with higher sensitivity than small-molecule indicators such as OGB-1 (*Chen et al., 2013*). The availability of multicolored variants in the GECO family and the RCaMP series allowed for simultaneous measurement of $Ca^{2+}$ dynamics in different cell populations in the same preparation, or in different subcellular compartments within the same cell (*Zhao et al., 2011*; *Akerboom et al., 2013*). These variants can be integrated with optogenetics to simultaneously evoke channel rhodopsin activity while monitoring localized $Ca^{2+}$ responses in independent spectral channels (*Akerboom et al., 2013*). Moreover, individual GECIs can be tagged onto membrane $Ca^{2+}$ channels to directly measure $Ca^{2+}$ influx through the target channel of interest, enabling optical recording of single channel activity without the need for technique-intensive patch clamping (*Dynes et al., 2016*).

Another advantage of GECIs is their capability to be incorporated into transgenic organisms. Although several GECI-expressing transgenic mouse lines have already been reported, many of these studies used older variants of GECIs that are expressed only in selected tissues (*Hasan et al., 2004*; *Ji et al., 2004*; *Tallini et al., 2006*; *Heim et al., 2007*). The Ai38 mouse line overcomes these issues by combining GCaMP3 with a robust and flexible Cre/lox system for selective expression in

specific cell populations (*Zariwala et al., 2012*). Based on a series of Cre-responder lines designed for characterization of the whole mouse brain (*Madisen et al., 2010*), the Ai38 mouse line contains GCaMP3 targeted to the Rosa26 locus but requires Cre recombinase for expression. By crossing Ai38 with various Cre mouse lines, GCaMP3 can be selectively expressed in specific cell populations. Thus, target cells may be endogenously labeled without invasive procedures, avoiding potential off-target side effects reported in GECI transgenic lines with global expression (*Direnberger et al., 2012*). The newly released PC::G5-tdT mouse line provides improved functionality by targeting a Cre-dependent GCaMP5G-IRES-tdTomato transgenic cassette to the *Polr2a* locus (*Gee et al., 2014*). However, in the PC::G5-tdT mouse line, GCaMP5G and tdTomato are expressed individually, and localize to different cell compartments. Moreover, because expression of tdTomato is driven by an internal ribosomal entry site, the expression level is highly variable and weaker than GCaMP5G, limiting identification of positive cells and preventing accurate ratiometric measurements.

Although single fluorescent protein-based indicators have high brightness and fast response kinetics, as non-ratiometric probes they are problematic for $Ca^{2+}$ imaging in motile cells where fluorescence changes resulting from movement may be indistinguishable from actual changes in $Ca^{2+}$ levels. Here, we introduce a novel genetically encoded $Ca^{2+}$ indicator - that we christen 'Salsa6f' - by fusing green GCaMP6f to the $Ca^{2+}$-insensitive red fluorescent protein tdTomato. This probe enables true ratiometric imaging, in conjunction with the high dynamic range of GCaMP6. We further describe the generation of a transgenic mouse enabling Salsa6f expression in a tissue-specific manner, and demonstrate its utility for imaging T lymphocytes in vitro and in vivo.

## Results

### A novel ratiometric genetically encoded $Ca^{2+}$ indicator, Salsa6f

In order to develop a better tool to monitor $Ca^{2+}$ signaling in T cells both in vivo and in vitro, we first evaluated the latest generation of genetically encoded $Ca^{2+}$ indicators (GECIs) (*Zhao et al., 2011*; *Chen et al., 2013*). We transiently expressed and screened a variety of single fluorescent protein-based GECIs in HEK 293A cells (*Figure 1A*), and selected GCaMP6f based on fluorescence intensity, dynamic range, and $Ca^{2+}$ affinity suitable for detecting a spectrum of cytosolic $Ca^{2+}$ signals ($K_d$ = 375 nM). To enable cell tracking even when basal $Ca^{2+}$ levels evoke little GCaMP6f fluorescence, we fused GCaMP6f to the $Ca^{2+}$-insensitive red fluorescent protein tdTomato, chosen for its photostability and efficient two-photon excitation (*Drobizhev et al., 2011*). A V5 epitope tag (*Lobbestael E et al., 2010*) served to link tdTomato to GCaMP6f (*Figure 1B*). The resultant ratiometric fusion indicator, coined 'Salsa6f' for the combination of red tdTomato with the green GCaMP6f, was readily expressed by transfection into HEK 293A cells and human T cells. Salsa6f exhibited a ten-fold dynamic range, with a brightness comparable to GCaMP6f alone (*Figure 1A,C*). For two-photon microscopy, both components of Salsa6f can be visualized by femtosecond excitation at 900 nm (*Figure 1D*). GCaMP6f produces increased green fluorescence during elevations in cytosolic $Ca^{2+}$, while tdTomato provides a stable red fluorescence that facilitates cell tracking and allows for ratiometric $Ca^{2+}$ imaging (*Figure 1D*; *Video 1*). Salsa6f is excluded from the nucleus, ensuring accurate measurement of cytosolic $Ca^{2+}$ fluctuations (*Figure 1D,E*). When expressed by transfection in human T cells, Salsa6f reported $Ca^{2+}$ oscillations induced by immobilized αCD3/28 antibodies with a high signal to noise ratio and time resolution (*Figure 1E,F*).

### Generation of Salsa6f transgenic reporter mice and validation in immune cells

Guided by the transgenic targeting strategy for the Ai38 mouse line (*Zariwala et al., 2012*), we inserted Salsa6f into a Gt(ROSA)26Sor5'-pCAG-FRT-LSL-Salsa6f-WPRE-bGHpA-AttB-FRT-NeoR-AttP-Gt(ROSA)26Sor3' cassette, then targeted it to the Rosa26 locus in JM8.N4 mouse embryonic stem (ES) cells (*Figure 2A*). Cells positive for the allele *Gt(ROSA)26Sor*^pCAG-FRT-LSL-Salsa6f-WPRE-bGHpA-AttB-FRT-NeoR-AttP^ were selected by neomycin resistance, and correctly targeted clones were screened by Southern blot (*Figure 2B*), then injected into C57BL/6J blastocysts for implantation. Chimeric pups carrying the Salsa6f transgene were identified by PCR screening for the *Nnt* gene, as the initial JM8.N4 ES cells were *Nnt*^+/+^ while the C57BL/6J blastocysts were *Nnt*^-/-^ (*Figure 2C*). Positive chimeras were bred to R26ΦC31o mice to remove the neomycin resistance gene and to produce LSL-

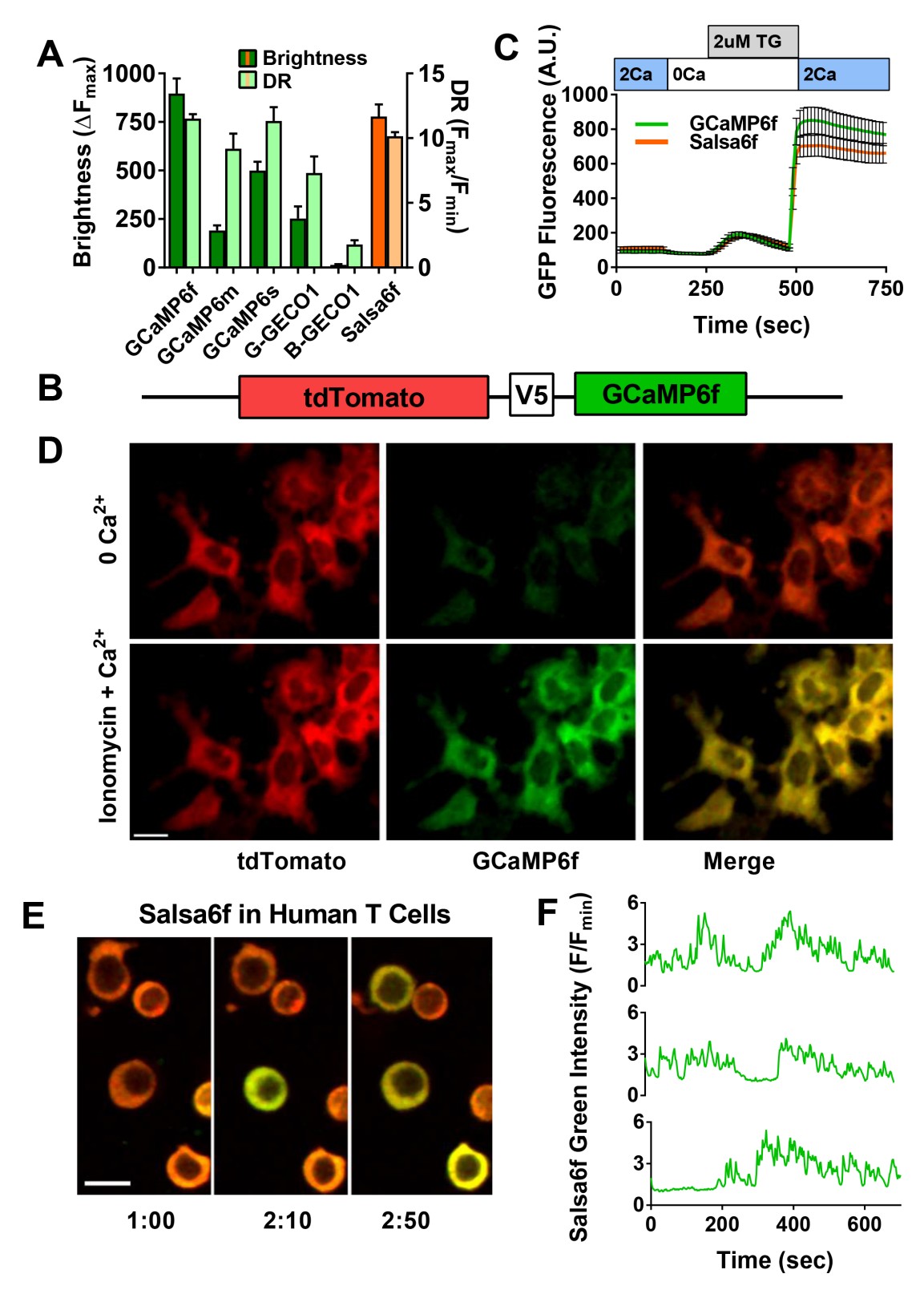

**Figure 1.** Design of novel tdTomato-V5-GCaMP6f fusion probe 'Salsa6f' and characterization in living cells. (**A**) Several genetically encoded $Ca^{2+}$ indicators were screened in vitro in HEK 293A cells, by co-transfecting with Orai1/STIM1 and measuring $Ca^{2+}$ influx after thapsigargin-induced store depletion. Bars indicate maximum change in fluorescence intensity (dark) and dynamic range (DR: light) with Salsa6f shown in orange bars on right; n > 30 cells per probe, from two different transfections, error bars indicate SEM. (**B**) Diagram of Salsa6f construct used in transfection. (**C**) Averaged

*Figure 1 continued on next page*

*Figure 1 continued*

thapsigargin-induced Ca$^{2+}$ entry, measured by change in green fluorescence, in GCaMP6f- (green, 11.5 ± 0.3, n = 63) or Salsa6f- (orange, 10.2 ± 0.3, n = 78) transfected HEK cells; data from two different transfections, error bars indicate SEM. (D) Two-photon images of Salsa6f co-transfected in HEK cells with Orai1/STIM1, showing red (tdTomato), green (GCaMP6f), and merged channels, at baseline in 0 mM extracellular Ca$^{2+}$ (top) and after maximum stimulation with 2 µM ionomycin in 2 mM extracellular Ca$^{2+}$ (bottom); scale bar = 20 µm; see *Video 1*; data are representative of at least three different experiments. (E) Confocal time lapse microscopy of human Cd4$^+$ T cells previously transfected with Salsa6f and then activated for 2 days on plate-bound αCd3/28 antibodies; time = min:s, scale bar = 10 µm. (F) Representative traces of green fluorescence intensity from individual activated human T cells transfected with Salsa6f. Data are representative of at least three different experiments.

DOI: https://doi.org/10.7554/eLife.32417.003

Salsa6f F1 founders that are heterozygotic for the *Gt(ROSA)26Sor*$^{pCAG-FRT-LSL-Salsa6f-WPRE-bGHpA-AttB/P}$ allele, then further bred to generate homozygotic mice which we term LSL-Salsa6f (Hom).

LSL-Salsa6f (Hom) mice were bred to *Cd4*$^{Cre+/+}$ mice to obtain reporter mice heterozygous for Salsa6f, designated as Cd4-Salsa6f (Het) mice from here on, that selectively express Salsa6f in T cells (*Figure 3A*). Mice homozygous for Salsa6f are designated as Cd4-Salsa6f (Hom). Salsa6f was detected by tdTomato fluorescence on flow cytometry. 88% of these Salsa6f$^+$ cells in thymus were double positive for Cd4 and Cd8 (*Figure 3B*). This is due to the double-positive stage during development, in which developing thymocytes will express both Cd4 and Cd8 before undergoing positive and negative selection to become either mature Cd4$^+$ or Cd8$^+$ T cells. Salsa6f was readily detected by the red tdTomato signal in cells from spleen (40%), lymph node (57%), and thymus (93%) (*Figure 3C*). As expected, double positive cells were not detected in the spleen (*Figure 3D*). More than 98% of Cd4$^+$ and Cd8$^+$ T cells from these reporter mice were positive for Salsa6f. Salsa6f was also detected in 5% of Cd19$^+$ cells and 3% of Cd11b$^+$ cells (*Figure 3E*). A small fraction of B cells express Cd4 mRNA, which may explain the presence of Salsa6f in Cd19$^+$ cells (*Zhang and Henderson, 1994*). Cd11b$^+$ cells positive for Salsa6f may be splenic resident dendritic cells that also express Cd4 (*Vremec et al., 2000*; *Turley et al., 2010*). The total number and relative frequencies of Cd4$^+$, Cd8$^+$, Cd19$^+$, and Cd11b$^+$ cells were similar to the *Cd4*$^{Cre}$ controls (*Figure 3F,G*).

To determine whether expression of Salsa6f might affect functional responses downstream of Ca$^{2+}$ signaling in T cells, we first purified Cd4$^+$ T cells and monitored cell proliferation in vitro during TCR engagement of αCd3 and co-stimulating αCd28 antibodies attached to activating beads. Both hetero and homozygotic Salsa6f-expressing Cd4$^+$ T cells proliferated similar to the *Cd4*$^{Cre}$ controls (*Figure 4A,B*). To further probe functional responses, we differentiated naive Cd4$^+$ T cells using polarizing cytokine stimuli to generate Th1, Th17 and induced regulatory T cells (iTregs). Salsa6f-positive naive Cd4$^+$ T cells from both Cd4-Salsa6f (Het) and Cd4-Salsa6f (Hom) mice readily differentiated into various helper T cell subtypes similar to the *Cd4*$^{Cre}$ controls (*Figure 4C–E* and *Figure 4—figure supplement 1*). In addition, as described in the companion paper, adoptively transferred Salsa6f-positive cells readily homed to lymph nodes and exhibited normal motility characteristics (*Dong et al., 2017*). In summary, our results demonstrate normal T-cell functions of Salsa6f-expressing T lymphocytes with respect to development, cellular phenotype, cell proliferation, differentiation, homing, and motility.

## Single-cell ratiometric Ca$^{2+}$ measurement in Cd4-Salsa6f reporter mice

Cd4$^+$ T cells were purified from Cd4-Salsa6f (Het) reporter mice, stimulated with plate-bound αCd3/28 antibodies for 2 days, and imaged by confocal microscopy while still in contact with immobilized antibodies (*Figure 5A*, *Video 2*). Activated Cd4$^+$ T cells expressing Salsa6f exhibited stable red fluorescence and wide fluctuations in green fluorescence due to Ca$^{2+}$ oscillations

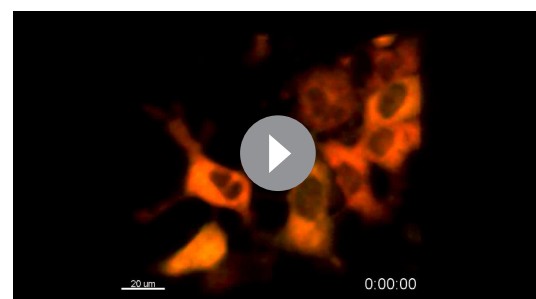

**Video 1.** Calcium readout of Salsa6f probe in HEK cells. HEK 293A cells transfected with Salsa6f, first washed with 0 mM Ca$^{2+}$ followed by 2 µM ionomycin in 2 mM Ca$^{2+}$; scale bar = 20 µm, time shown in hr:min:s. Images were acquired at 15 s interval and played back at 15 frames per second. This video corresponds to *Figure 1D*.
DOI: https://doi.org/10.7554/eLife.32417.004

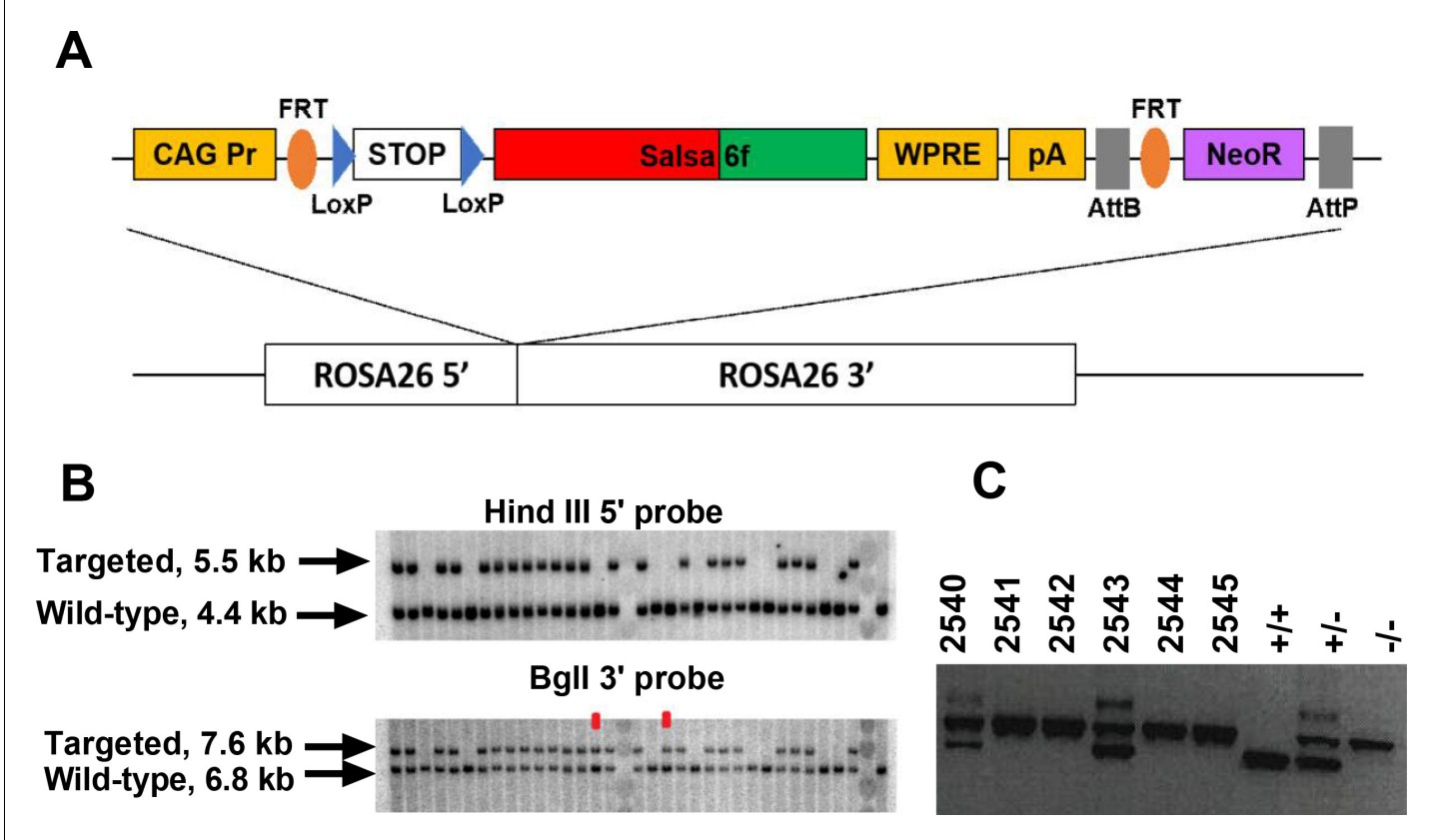

**Figure 2.** Generation of a Salsa6f transgenic mouse line targeted to the Rosa26 locus. (A) Transgenic targeting vector for Salsa6f, inserted between Rosa26 homology arms and electroporated into embryonic stem cells. CAG Pr: cytomegalovirus early enhancer/chicken β-actin promoter; Salsa6f: tdTomato-V5-GCaMP6f; FRT, LoxP, AttB, AttP: recombinase sites; WPRE: woodchuck hepatitis virus post-transcriptional regulatory element; pA: bovine growth hormone polyadenylation sequence; NeoR: neomycin resistance gene. (B) Correctly targeted ES cells were screened by Southern blot after HindIII digest for the 5' end (top) or BglI digest for the 3' end (bottom). The two clones marked in red failed to integrate at the 5' end. (C) PCR screening for chimeras based on presence of the Nnt mutation, present only in JM8.N4 ES cells but not in the C57BL/6J blastocyst donors. 2540 and 2543 are chimeras. Control lanes on the right are wild type ($Nnt^{+}/^{+}$), heterozygous ($Nnt^{+/-}$), or homozygous mutant ($Nnt^{-/-}$).
DOI: https://doi.org/10.7554/eLife.32417.005

resulting from T-cell receptor engagement (*Figure 5B*). Despite variability in total fluorescence between cells due to individual differences in cell size, the basal and peak green/red Salsa6f ratios (referred from now on as G/R ratio for GCaMP6f/tdTomato intensity) were comparable between cells and showed up to six-fold increases during peaks in $Ca^{2+}$ fluctuations (*Figure 5C*). Flow cytometric analysis of Salsa6f mouse T cells revealed a 13-fold increase in G/R ratio, by pretreatment with ionomycin in $Ca^{2+}$-free medium to deplete cytosolic $Ca^{2+}$ followed by addback of extracellular $Ca^{2+}$, further emphasizing the high dynamic range of Salsa6f (*Figure 5D*). Finally, to test if increasing the genetic dosage can improve the brightness of Salsa6f, we compared $Cd4^{+}$ T cells from Cd4-Salsa6f (Het) and Cd4-Salsa6f (Hom) mice. T cells from homozygous mice with two allelic copies of the Salsa6f reporter cassette exhibited almost a two-fold increase in tdTomato fluorescence compared to heterozygous mice (*Figure 5E*), allowing for genetic control of Salsa6f expression level when brightness is an issue.

## Cytosolic localization and calibration of Salsa6f in transgenic T lymphocytes

We first examined the localization of Salsa6f in naïve $Cd4^{+}$ T cells isolated from Cd4- Salsa6f (Het) mice and in $Cd4^{+}$ T cells activated for 2 days on plate-bound αCd3/28. Line scans of the confocal images of cells plated on poly-L-lysine-coated coverslips showed that Salsa6f is primarily localized to the cytoplasm and is excluded from the nucleus (*Figure 6A,B*). Increasing the cytosolic $Ca^{2+}$ levels

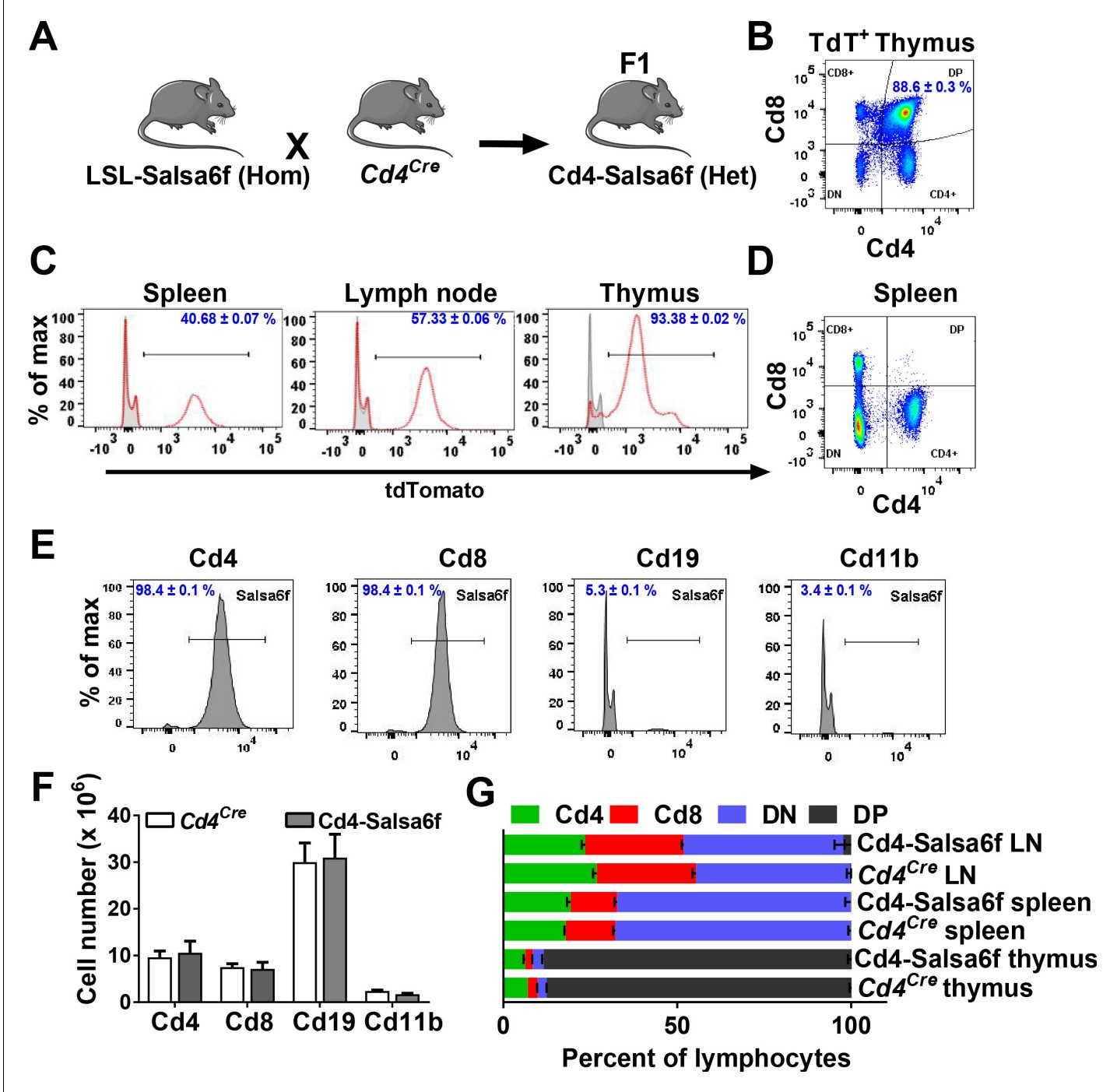

**Figure 3.** Cd4-Salsa6f mice show normal immune cell development and expression. (A) Experimental design to target expression of Salsa6f in Cd4 cells. (B) Cd4, Cd8 and double-positive cells gated on tdTomato (Salsa6f+ cells) from thymus. (C) Histograms showing percent of Salsa6f+ cells in spleen, LN, and thymus. (D) Cd4, Cd8, and double positive cells from spleen, gated on tdTomato (Salsa6f+ cells). (E) Histograms showing percent of Salsa6f+ cells within Cd4, Cd8, Cd19, Cd11b populations from spleen. (F) Total number of Cd4, Cd8, Cd19, Cd11b cells in the spleen of Cd4-Salsa6f (Het) mice and *Cd4Cre* mice (n = 6 mice). (G) Relative percentages of Cd4, Cd8, Cd19, Cd11b cells in thymus, lymph nodes, and spleen of Cd4-Salsa6f mice and *Cd4Cre* mice (n = 6).

DOI: https://doi.org/10.7554/eLife.32417.006

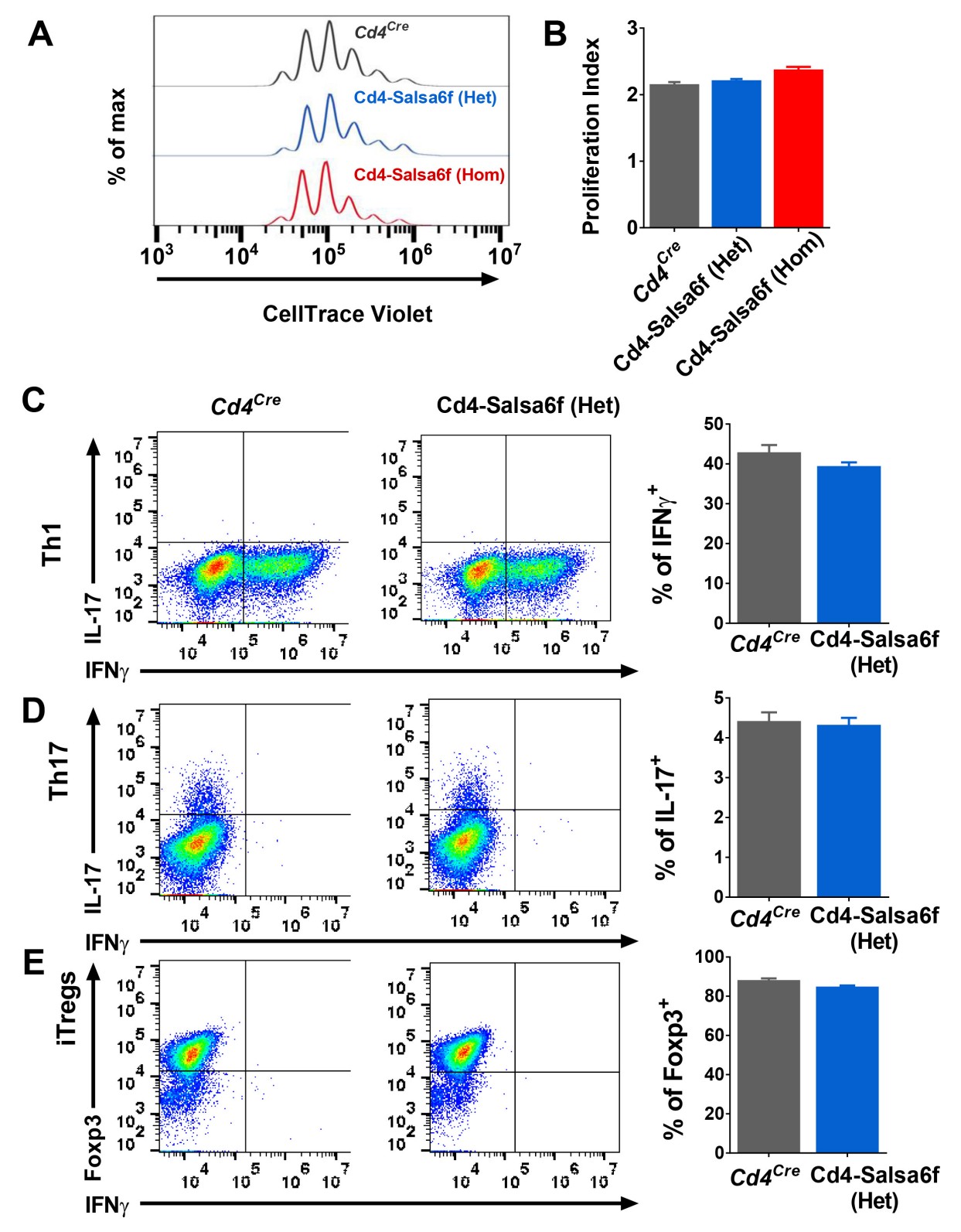

**Figure 4.** Functional responses of Cd4-Salsa6f T cells in vitro. (**A**) Representative histogram showing cell trace violet (CTV) dilution in *Cd4^Cre* (dark grey), Cd4-Salsa6f (Het) (blue), and Cd4-Salsa6f (Hom) (red) T cells at 72 hr following stimulation with αCd3/28 Dynabeads (1:1 ratio). (**B**) Proliferation index measured on CTV dilution curves (n = 8). (**C–E**) Dot plots showing differentiation of naive T cells from *Cd4^Cre* and Cd4-Salsa6f (Het) mice into Th1 cells
*Figure 4 continued on next page*

*Figure 4 continued*

(C), Th17 cells (D) and iTregs (E) after 6 days (n = 4 mice). Right panels show average percentages of IFNγ⁺ cells (C), IL-17⁺ cells (D) and Foxp3⁺ cells (E).

DOI: https://doi.org/10.7554/eLife.32417.007

The following figure supplement is available for figure 4:

**Figure supplement 1.** Differentiation of Cd4-Salsa6f (Hom) T cells into Th1 and iTregs.

DOI: https://doi.org/10.7554/eLife.32417.008

using thapsigargin (TG) in 2 mM $Ca^{2+}$ Ringer's solution caused a selective increase in the GCaMP6f signal, without altering the localization of Salsa6f probe. In contrast, the small-molecule dye $Ca^{2+}$ indicators fluo-4 or fura-2 loaded into $Cd4^+$ T cells from $Cd4^{Cre}$ mice are distributed throughout the cell, including the nucleus (*Figure 6C* and data not shown). A different transgenic mouse, PC::G5-tdT utilizes an internal ribosomal entry site to express both tdTomato and GCaMP5G as separate proteins that localize differently (*Gee et al., 2014*), with tdTomato distributed throughout the cell including the nucleus and GCaMP5G predominantly in the cytosol (*Figure 6—figure supplement 1*). In contrast, our tandem probe, Salsa6f, results in both red and green fluorescent proteins co-localized in the cytosol, allowing true ratiometric $Ca^{2+}$ imaging and facilitating tracking of cells.

Overlap of GCaMP6f emission and tdTomato excitation spectra raises the possibility of FRET (Forster resonance energy transfer) which would decrease G/R ratio and reduce dynamic range. If there were significant FRET, we would expect acceptor (tdTomato) signal to increase with increase in donor (GCaMP6f) intensity, as shown for GECI probes utilizing GCaMP and mCherry dependent on the rigidity of the linker used (*Cho et al., 2017*). We treated Salsa6f-positive T cells with Ringer's buffer containing Ionomycin and different concentrations of $Ca^{2+}$ and measured the steady-state GCaMP6f and tdTomato fluorescence signals. The GCaMP6f signal increased as expected with increasing $Ca^{2+}$ concentrations, but the tdTomato signal remained unchanged (*Figure 6D*). This suggests that the V5 epitope tag is effective in reducing FRET between GCaMP6f and tdTomato which otherwise may compromise probe performance.

We next characterized the in situ $Ca^{2+}$ affinity of Salsa6f in $Cd4^+$ T cells isolated from Cd4-Salsa6f (Het) mice, using fura-2 as a calibration standard. Fura-2 has an in situ $K_d$ ~225 nM at 25° C (*Lewis and Cahalan, 1989*), which is somewhat lower than in vitro $K_d$ values reported for GCaMP6f (*Chen et al., 2013*; *Badura et al., 2014*). We used time-lapse imaging to record G/R ratio in response to ionomycin applied with stepwise increases in the external $Ca^{2+}$ concentration and, using identical protocols, compared it to fura-2 $Ca^{2+}$ signals in control $Cd4^+$ T cells from $Cd4^{Cre}$ mice (*Figure 6E,F*). To facilitate comparison, fura-2 and Salsa6f ratios were normalized with $R_{min} = 0$ and $R_{max} = 1$. Salsa6f responded with faster rise and decay kinetics than fura-2 to progressive increases in cytosolic $Ca^{2+}$ levels, especially at lower external $Ca^{2+}$ concentrations. Salsa6f responses also saturated at lower cytosolic $Ca^{2+}$ levels than fura-2 responses. This is not altogether surprising given that genetically encoded $Ca^{2+}$ indicators have been reported to have a steeper Hill coefficient than chemical indicators (*Badura et al., 2014*). Assuming that Cd4-Salsa6f T cells reach similar $Ca^{2+}$ levels as control $Cd4^{Cre}$ cells, we plotted peak and the steady state G/R ratios against the cytosolic $Ca^{2+}$ concentrations obtained from the fura-2 experiment which yielded an estimated in situ $K_d$ for Salsa6f in the range of 160–300 nM (*Figure 6G*). Based on these results, we conclude that Salsa6f is sensitive in detecting cytosolic $Ca^{2+}$ from 100 nM - 2 µM, which is in the range of physiological cytosolic $Ca^{2+}$ signals.

## T-cell $Ca^{2+}$ signaling in response to $Ca^{2+}$ store depletion, T-cell receptor engagement, and mechanical stimulation

TCR engagement activates a canonical $Ca^{2+}$ signaling pathway in T cells, characterized by $IP_3$-induced $Ca^{2+}$ release from the endoplasmic reticulum, leading to store-operated $Ca^{2+}$ entry (SOCE) through Orai1 channels (*Cahalan and Chandy, 2009*; *Prakriya and Lewis, 2015*). Past studies on T cell $Ca^{2+}$ signaling have largely relied on indicators like fura-2 and fluo-4 which have the potential drawback of being distributed into the nucleus, thus confounding the measurement of pure cytoplasmic $Ca^{2+}$ signals, a problem particularly notable in T cells with their large nuclear to cytoplasmic volume ratio. Salsa6f, with its high dynamic range, ratiometric readout and targeted localization in the cytosol, is thus well suited to record physiological cytosolic $Ca^{2+}$ signals. To this end, we

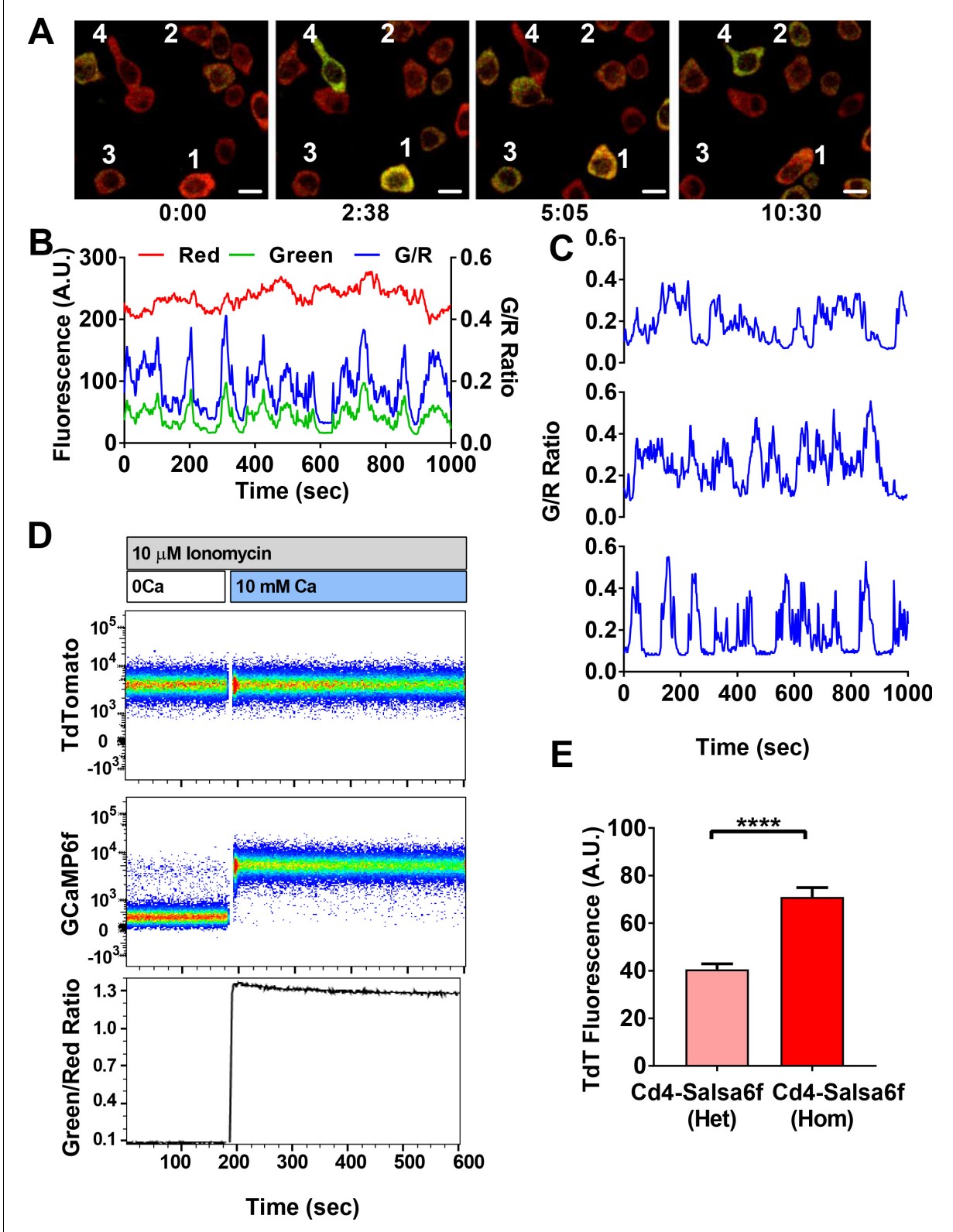

**Figure 5.** Single-cell Salsa6f calcium signals in T cells. (A) Confocal images of Ca$^{2+}$ signals in activating Cd4$^+$ T cells from Cd4-Salsa6f (Het) mice, after two day stimulation on plate bound αCd3/28 antibody, showing merged green (GCaMP6f) and red (tdTomato) channels; time = min:sec; scale bar = 10 μm. (B) Representative traces from cell #3 in (A), showing cell-wide fluorescence intensity changes in GCaMP6f (green), tdTomato (red), and green/red ratio (G/R, blue). (C) G/R ratios for cells 1, 2, and 4 from (A). (D) Dynamic range of Salsa6f in resting Cd4 T cells, measured as green/red fluorescence

*Figure 5 continued on next page*

*Figure 5 continued*

ratio by flow cytometry. Cells were pre-treated with 10 µM ionomycin in Ca$^{2+}$-free solution (white bar), followed by re-addition of 10 mM Ca$^{2+}$ (blue bar). (**E**) Averaged tdTomato fluorescence in resting T cells from heterozygous Cd4-Salsa6f compared to homozygotic Cd4-Salsa6f mice.

DOI: https://doi.org/10.7554/eLife.32417.009

recorded Ca$^{2+}$ signals from 2-day-activated Cd4$^+$ T cells from Cd4-Salsa6f (Het) mice in response to a variety of stimuli.

To study SOCE more directly, we depleted ER Ca$^{2+}$ stores in activated and in naïve T cells by applying TG in Ca$^{2+}$-free solution. We observed a small but sharp initial peak indicating ER store release followed by a sustained Ca$^{2+}$ signal upon restoring Ca$^{2+}$ to the external bath, indicative of SOCE (*Figure 7A*, *Video 3* and *Figure 7—figure supplement 1A*). Almost all cells responded to this supra-physiological stimulus (*Figure 7A*, right panel). In contrast, T cells plated on αCd3/28 to activate TCR-induced signaling showed heterogeneous responses, with some exhibiting asynchronous Ca$^{2+}$ oscillations of varying frequencies and durations whereas others failed to respond (*Figure 7B*, *Video 4*). Past studies have attributed these Ca$^{2+}$ oscillations to SOCE from repetitive opening and closing of Orai1 channels allowing Ca$^{2+}$ to enter T cells in a periodic and asynchronous manner (*Lewis and Cahalan, 1989*; *Dolmetsch and Lewis, 1994*), unlike the sustained activation with TG treatment. Cells plated on αCd3 alone also showed rhythmic oscillatory Ca$^{2+}$ signals; however, the percentage of responding cells was significantly lower than with αCd3/28, resulting in a lower average signal compared with αCd3/28 stimulation (*Figure 7C*, *Video 5*) These results suggest that co-stimulatory signaling through Cd28 enhances the response to TCR activated Ca$^{2+}$ signals, in alignment with previous observations (*Chen and Flies, 2013*).

Finally, we focused on a novel Ca$^{2+}$ signaling pathway involving mechanosensitive Ca$^{2+}$-permeable channels. We examined activation of mechanosensitive channels in Salsa6f$^+$ T cells that were plated on αCd3/28-coated glass coverslips, by flowing external solution rapidly past the cells. Perfusion produced a transient rise in the cytosolic Ca$^{2+}$ signal, likely a result of flow shear stress (*Figure 7D* and *Figure 7—figure supplement 1B*). The Piezo family of mechanosensitive channels plays a vital role in cell motility and development (*Nourse and Pathak, 2017*), but it is not known whether these channels are expressed and play a role in immune cell function. Perfusion of medium including Yoda1, a selective small molecule activator of Piezo1 (*Syeda et al., 2015*), resulted in robust Ca$^{2+}$ signals in activated and naive T cells that were larger and more sustained than those activated by perfusion of solvent or media alone (*Figure 7D* and *Figure 7—figure supplement 1C*). In summary, we show Ca$^{2+}$ signals in T cells in response to an activator of Piezo1 channels. These results illustrate the utility of Salsa6f for screening agents that modulate Ca$^{2+}$ signaling in T cells and open the possibility for further exploration of the functional role of Piezo1 channels in T cell function.

## TCR-induced Ca$^{2+}$ signaling in helper T cell subsets

We also monitored Ca$^{2+}$ signaling in response to TCR activation by αCd3/28 in various subsets of T cells from Cd4-Salsa6f (Het) mice, including naive T cells, Th17 cells and iTregs (*Figure 8A–C*). All subtypes of T cells responded to plate-bound stimulation of αCd3/28 with oscillatory changes in their cytosolic Ca$^{2+}$ levels, very similar to the Ca$^{2+}$ responses in 2-day-activated T cells illustrated in *Figure 7B*. The responses were heterogeneous, with some cells showing multiple peaks of varying durations and amplitudes, occasional sustained signals and other cells failing to respond. Whereas the overall average responses were not very different between the three subtypes examined, some individual Th17 cells and iTregs showed higher amplitude signals than any naive T cells, but with a greater percentage of

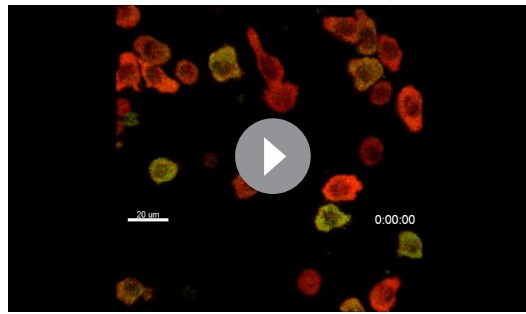

**Video 2.** Single-cell readout of activation in transgenic T cells by Salsa6f. Cd4 T cells from Cd4-Salsa6f (Het) mice were plated on activating surface coated with anti-Cd3/Cd28. Images were acquired at 5 s interval and played back at 15 frames per second. This video corresponds to *Figure 5A*.

DOI: https://doi.org/10.7554/eLife.32417.010

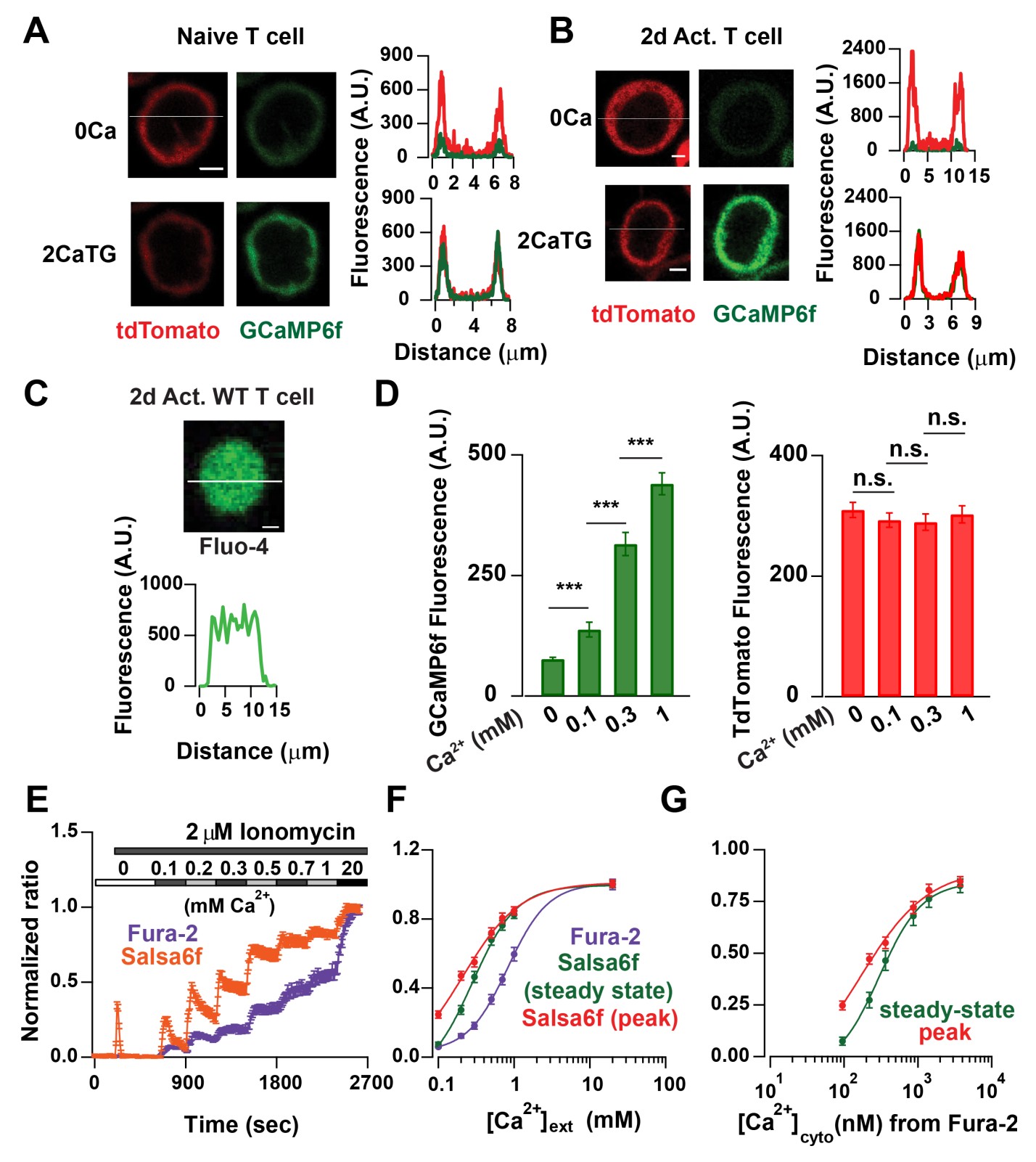

**Figure 6.** Probe characterization and calibration of $[Ca^{2+}]$ in Salsa6f T cells. (**A**) Confocal images of a naive T cell from a Cd4-Salsa6f (Het) mouse. Upper panel: tdTomato (left) and GCaMP6f (right) fluorescence intensity in $Ca^{2+}$-free Ringer solution. Lower panel: same cell treated with 2 µM thapsigargin (TG) in Ringer solution containing 2 mM $Ca^{2+}$. Line scans for each condition are shown adjacent to the images. Scale bar = 2 µm for **A–C**). (**B**) Corresponding confocal images and line scans of Salsa6f localization in a 2-day activated Cd4+ T-cell from Cd4-Salsa6f (Het) mouse. (**C**) Confocal image

*Figure 6 continued on next page*

*Figure 6 continued*

of a Fluo-4 (5 µM)-loaded Cd4$^+$ T cell from *Cd4$^{Cre}$* mouse. (D) Average GCaMP6f and tdTomato intensities in 2-day activated Cd4$^+$ T cells treated with 2 µM ionomycin in Ca$^{2+}$-free buffer and in external Ringer solution containing 0.1, 0.3 and 1 mM Ca$^{2+}$. n = 36 cells, representative of three experiments. (E) Average 340/380 nm ratios in 2-day-activated and fura-2-loaded Cd4$^+$ T cells from *Cd4$^{Cre}$* mice (n = 59 cells) and G/R ratios in 2-day activated Cd4$^+$ T cells from Cd4-Slsa6f (Het) mice (n = 47 cells) treated identically with 2 µM ionomycin followed by graded increases of external Ca$^{2+}$ concentration as indicated. (F) Steady-state fura-2 and Salsa6f ratios recorded 300 s after solution application and peak Salsa6f ratio from **6E** plotted as a function of external Ca$^{2+}$ concentration. (G) Steady-state and peak Salsa6f ratios plotted as a function of cytosolic Ca$^{2+}$ concentrations calculated from the fura-2 experiment, assuming a fura-2 $K_d$ of 225 nM. The points were fit with a four parameter Hill equation to obtain the $K_d$ for Salsa6f, with the following parameters: Salsa6f steady-state: Hill coefficient = 1.49 ± 0.16; $K_d$ = 301 ± 24; Salsa6f peak: Hill coefficient = 0.93 ± 0.4; $K_d$ = 162 ± 48. Data are representative of three experiments.

DOI: https://doi.org/10.7554/eLife.32417.011

The following figure supplement is available for figure 6:

**Figure supplement 1.** Comparison of GECI localization in Cd4 T cells from Salsa6f mouse and PC::G5-tdT mouse.
DOI: https://doi.org/10.7554/eLife.32417.012

non-responding cells. The Cd4-Salsa6f mouse thus opens up new avenues to study the fundamental nature of Ca$^{2+}$ signals in T cell subsets generated in response to variety of stimuli, and to explore the relationship between patterns of Ca$^{2+}$ signals and specific downstream functions.

## Two-photon microscopy of Cd4-Salsa6f T cells in the lymph node

Using two-photon microscopy to image lymph nodes from Cd4-Salsa6f (Hom) mice under steady-state conditions, we observed sporadic localized elevations of green fluorescence indicative of intracellular Ca$^{2+}$ signaling (*Figure 9A,B*). Imaging conditions were optimized for single-wavelength excitation of both TdTomato and GCaMP6f components of Salsa6f (*Figure 9—figure supplement 1*). Some of the green Salsa6f signals were T cell-sized, and the pattern of red Salsa6f fluorescence observed is consistent with the exclusion of the Salsa6f indicator from the nucleus (*Figure 9C*). In addition, we observed numerous bright, transient green fluorescent signals which were much smaller in area (about 2 µm$^2$ in area) (*Figure 9B,D*; *Video 6*). We term these fluorescent transients 'sparkles', because during rapid playback of time-lapse image streams cells appear to sparkle.

Because T cells move rapidly and are not uniformly distributed in lymph nodes, we developed an image processing approach in order to minimize fluctuations in background fluorescence and in order to sensitively identify cell-wide Ca$^{2+}$ signals and sparkles (*Figure 9—figure supplement 2*). Based on the one-to-one correspondence of tdTomato and GCaMP6f, we estimated and subtracted

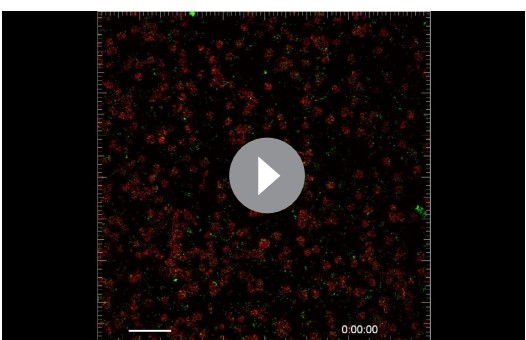

**Video 3.** T cell Ca$^{2+}$response to Ca$^{2+}$store depletion by thapsigargin (TG). Video of maximum intensity projection images of 2-day activated T cells from Cd4-Salsa6f (Het) mouse plated on poly-L-lysine. Scale bar = 20 µm, time shown in hr:min:s. 2 µM TG in Ca$^{2+}$-free Ringer's was added at 00:02:30 and 2 mM Ca$^{2+}$was added at 00:08:15. Time interval between frames is 5 s. Play back speed = 50 frames per second. This video corresponds to *Figure 7A*.
DOI: https://doi.org/10.7554/eLife.32417.015

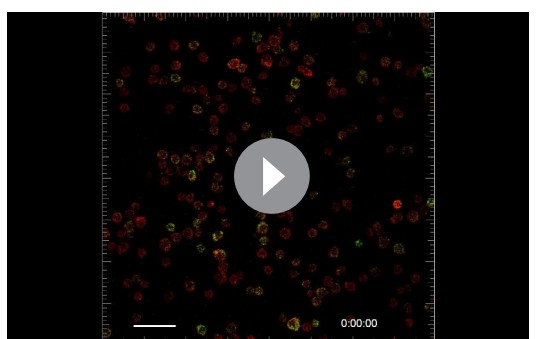

**Video 4.** Activated T cell Ca$^{2+}$responses to TCR stimulation. Video of maximum intensity projection images of 2-day activated T cells from Cd4-Salsa6f (Het) mouse plated on anti-Cd3/28-coated coverslip. Scale bar = 20 µm, time shown in hr:min:s. Time interval between frames is 5 s. Play back speed = 15 frames per second. Video corresponds to *Figure 7B*.
DOI: https://doi.org/10.7554/eLife.32417.016

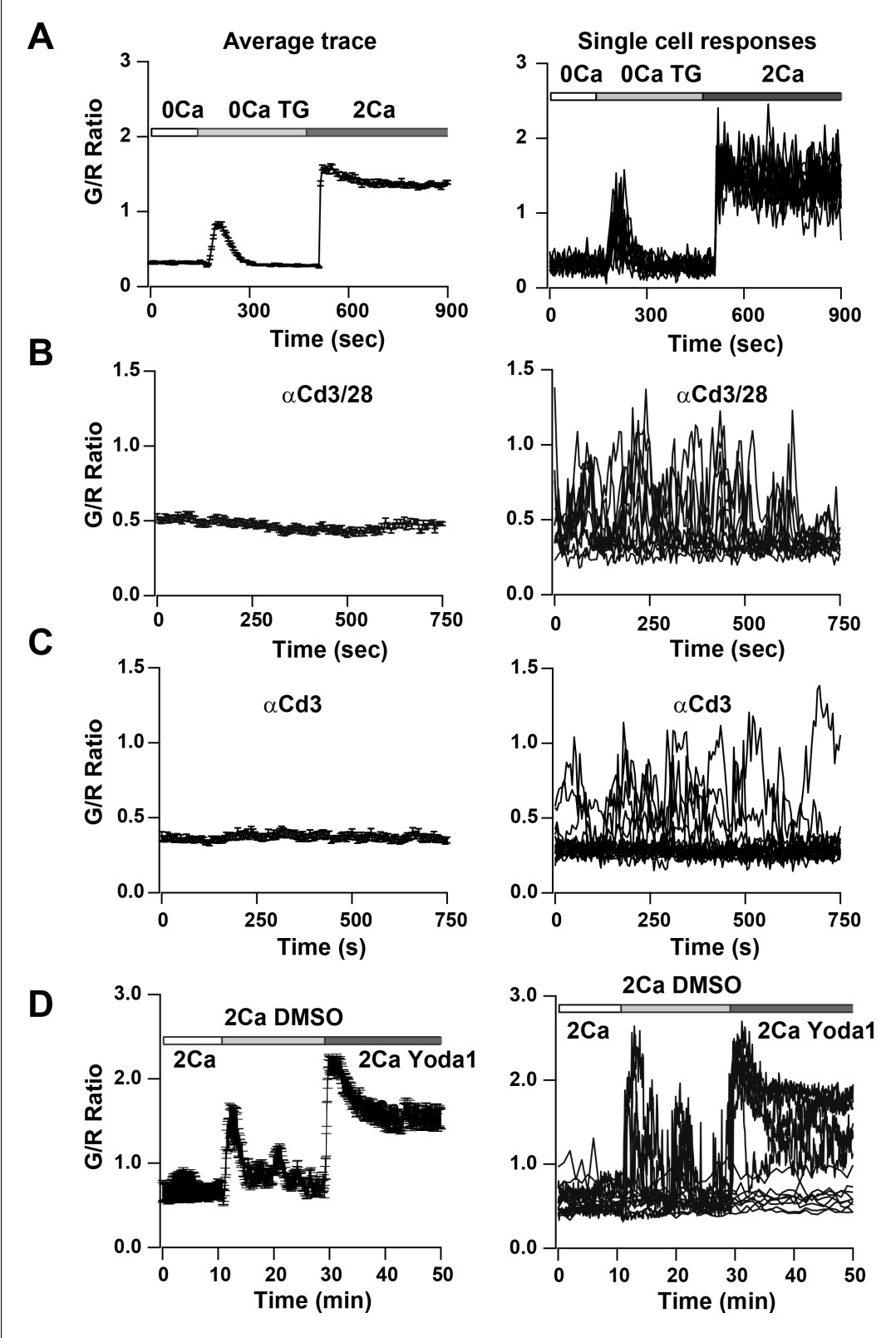

**Figure 7.** Ca$^{2+}$signals in two day activated Cd4$^+$ T cells from Cd4-Salsa6f (Het) mice in response to store-depletion, TCR stimulation and stimulation by Yoda1. In all panels, average Salsa6f G/R ratios are shown on the left, and representative single-cell traces are shown superimposed on right. Experiments were done in standard Ringer solution (**A**) or in RPMI containing 2% FCS and 2 mM Ca$^{2+}$ (**B–D**). (**A**) Store-operated Ca$^{2+}$ entry (SOCE) in Cd4$^+$ T cells (n = 86 cells), induced by depleting ER Ca$^{2+}$ stores with TG in Ca$^{2+}$-free buffer followed by re-addition of Ringer containing 2 mM Ca$^{2+}$. (**B**,

*Figure 7 continued on next page*

*Figure 7 continued*

C) Ca$^{2+}$ responses to TCR stimulation in T cells plated on coverslips coated with 1 µg/ml αCd3/Cd28 (B) or 1 µg/ml αCd3 alone (C) (n = 90 cells each). (D) Ca$^{2+}$ elevations during shear stress induced by solution exchange followed by the Piezo1 agonist Yoda1 (15 µM) in cells plated on αCd3/28 (n = 79 cells).

DOI: https://doi.org/10.7554/eLife.32417.013

The following figure supplement is available for figure 7:

**Figure supplement 1.** Store-operated Ca$^{2+}$entry and Piezo1 activation in T cells from Cd4-Salsa6f (Het) mice.

DOI: https://doi.org/10.7554/eLife.32417.014

out fluctuations in green fluorescence due to cell movement and distribution. After processing, sparkles were found to occur widely across the imaging field (*Figure 9E–G*), and many reached a characteristic maximum intensity (compare *Figure 9F and G*). The brightness of sparkles and the uniformity of background fluorescence allowed us to use a stringent threshold of signals exceeding 5.4 SD of the background noise level to systematically identify bright sparkles. Hundreds of events with a mean amplitude of 6.5 SD above background were observed in each 25 min imaging session (one image stack every five seconds), whereas fewer than one event exceeding our threshold was expected to occur from random noise fluctuations. Sparkles were more frequent than cell-wide transients (*Figure 9H*), extended over a median area of 1.9 µm$^2$ (n = 441 sparkles from three lymph nodes; *Figure 9I*), and usually lasted for only one or two consecutive frames (*Figure 9J*). Sparkle trace shape differs from that expected for autofluorescent cell processes drifting into the imaging field (*Figure 9K*). Taken together, these observations suggest that sparkles correspond to local Ca$^{2+}$ signals restricted to small subcellular domains of T cells migrating through the lymph node.

The high density of expressing fluorescent T cells in the lymph nodes of Cd4-Salsa6f (Hom) mice makes it difficult to visualize and study subcellular Ca$^{2+}$ signals within individual cells. We thus adoptively transferred Cd4-Salsa6f T cells into wild-type recipients so these cells could be viewed in isolation at low density. Small green fluorescence transients seen in lymph nodes after adoptive transfer were similar in intensity to transients seen in Cd4-Salsa6f (Hom) lymph nodes (*Figure 10A*). When small, bright, and brief green fluorescence signals were traced back, they were found to originate in red fluorescent Cd4-Salsa6f (Hom) T cells (*Figure 10B–M*). Cell movement was used to define the front and back of labeled T cells for mapping the subcellular location of Ca$^{2+}$ signals. Some sparkles, identified in processed green channel images, were found to selectively localize to the front or back of motile T cells in which both the cell front and back were clearly seen (*Figure 10D,E,J,K*). Green-red channel ratiometric images, enabled by Salsa6f labeling, confirmed differences in Ca$^{2+}$ concentration between front and back, although regions of elevated Ca$^{2+}$ were also observed flanking the nucleus (*Figure 10F,L*). Differences in green-red channel ratiometric pixel intensities between front and back were highly significant (*Figure 10G,M*; p<0.0001, Mann–Whitney test). Thus, motile T cells exhibit Ca$^{2+}$ signals that are largely restricted to subregions of the cytoplasm, and these Ca$^{2+}$ signals – sparkles – were identified with high signal-to-noise ratio by Salsa6f expression. In a companion paper (*Dong et al., 2017*), we use Salsa6f transgenic mice to consider the relationship between Ca$^{2+}$ signals, both cell-wide and local, and T-cell motility in the lymph node.

## Discussion

We introduce Salsa6f, a novel, ratiometric genetically-encoded Ca$^{2+}$ probe. Salsa6f is a fusion of the high-performing green fluorescent GECI GCaMP6f and the bright red fluorescent

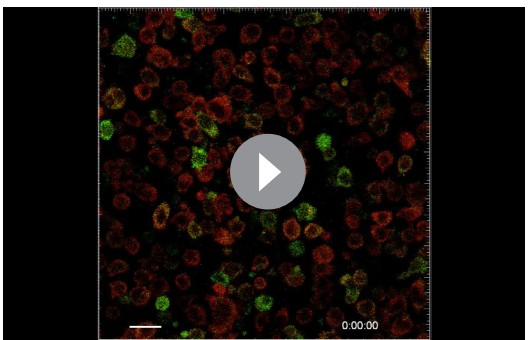

**Video 5.** T cell Ca$^{2+}$response to shear and Yoda1. Video of maximum intensity projection images of 2-day activated T cells from Cd4-Salsa6f (Het) mouse plated on anti-Cd3/28-coated coverslip. Scale bar = 20 µm, time shown in hr:min:s. Time interval between frames is 5 s. Play back speed = 200 frames per second. Medium was added at 00:15:00and Yoda1 was added at 00:35:00. Video corresponds to *Figure 7C*.

DOI: https://doi.org/10.7554/eLife.32417.017

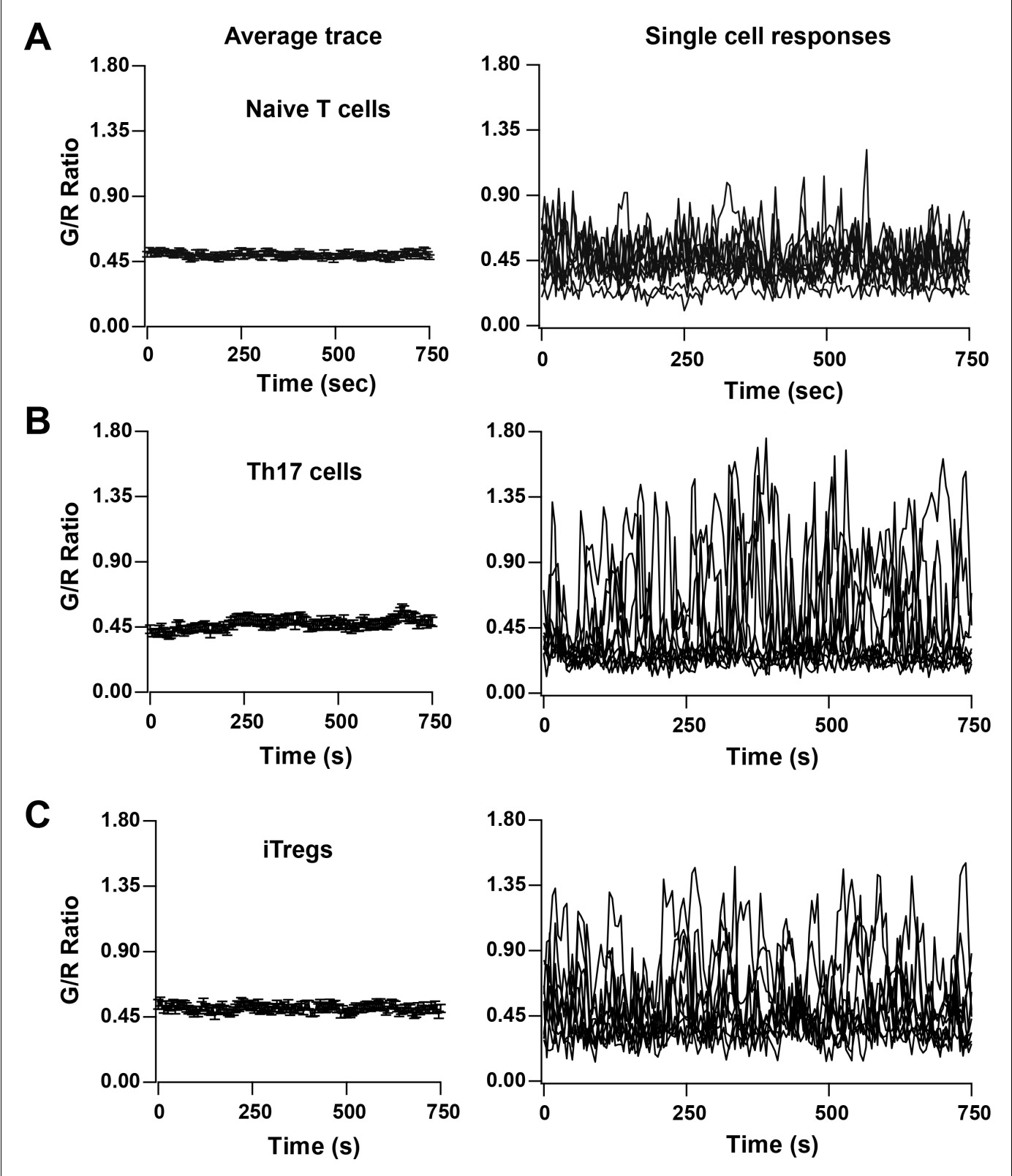

**Figure 8.** TCR induced Ca²⁺ signals in T cell subsets from Cd4-Salsa6f (Het) mice. Average (left) and representative single-cell Ca²⁺ traces (right) from confocal time-lapse microscopy showing changes in Salsa6f green/red (G/R) ratio in naive T cells (**A**), 5-day differentiated Th17 cells (**B**), and 5-day differentiated iTregs (**C**) plated on 1 µg/mL αCd3/28. (n = 90 cells from two to three experiments each).
DOI: https://doi.org/10.7554/eLife.32417.018

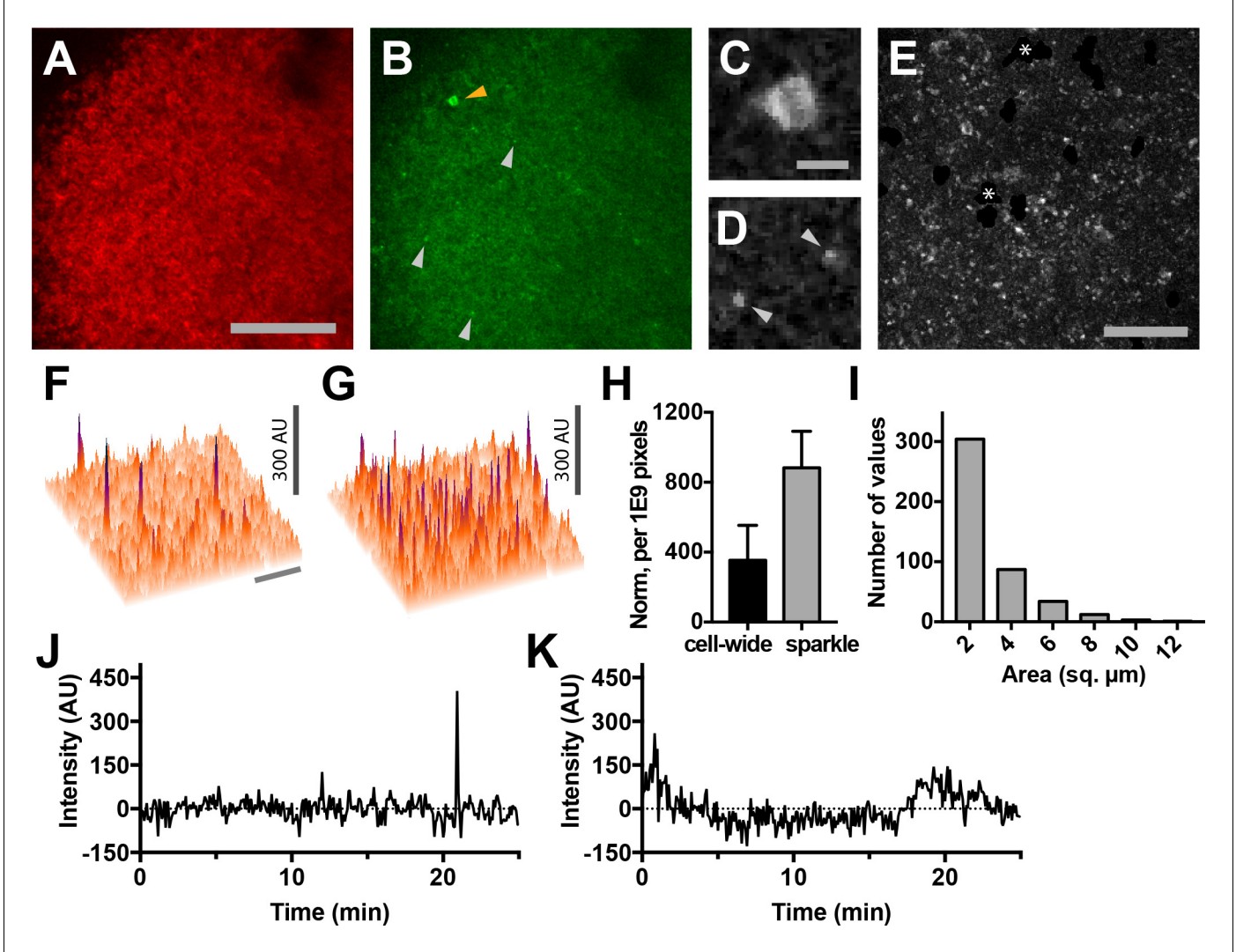

**Figure 9.** Lymph nodes from Cd4-Salsa6f (Hom) mice exhibit cell-wide and subcellular $Ca^{2+}$ signals. (**A**) Median filtered, maximum intensity projection of a red channel image from a single time point of an explanted lymph node from a Cd4-Salsa6f (Hom) mouse. (**B**) Green channel image corresponding to A). Orange arrowhead indicates cell-wide $Ca^{2+}$ signal and gray arrowheads indicate smaller, local transient $Ca^{2+}$ signals. (**C, D**) Enlargements of cell-wide (**C**) and local (**D**; gray arrowheads) $Ca^{2+}$ signals. Note the lower fluorescence intensity in the center of the cell in C due to exclusion of Salsa6f from the nucleus. (**E**) Maximum intensity projection of 214 green channel time points (every 11.5 s over 41 min) showing hundreds of small local $Ca^{2+}$ signals. Green channel image series was red channel subtracted and cropped from B. Asterisks indicate regions containing autofluorescent cells that have been cropped out. (**F, G**) Surface plot of maximum green channel intensity over two (**F**) and 50 (**G**) consecutive time points. Note the presence of four (**F**) and dozens (**G**) of small, discrete, high-intensity peaks of similar intensity. (**H**) Bar graph of relative frequencies of cell-wide and local $Ca^{2+}$ signals. (**I**) Frequency distribution of the area of local $Ca^{2+}$ signals. Scale bar in **A** is 100 μm (applies to **B**); scale bar in **C** is 10 μm (applies to **D**), scale bars in **E** and in **F** are 50 μm (applies to **G**). (**J**) Trace of fluorescence intensity over 25 min at the location of a transient subcellular $Ca^{2+}$ signal (one time point every 5 s). (**K**) Trace of fluorescence intensity of a putative cell process from an autofluorescent cell drifting in the image field.
DOI: https://doi.org/10.7554/eLife.32417.019

The following figure supplements are available for figure 9:

**Figure supplement 1.** Imaging lymph nodes of Cd4-Salsa6f homozygous mice.
DOI: https://doi.org/10.7554/eLife.32417.020
**Figure supplement 2.** Subtraction of red channel fluorescence improves detection of Salsa6f $Ca^{2+}$ signals.
DOI: https://doi.org/10.7554/eLife.32417.021

tdTomato. This simple modification imparts powerful capabilities, which facilitate tracking cells in the absence of $Ca^{2+}$ signaling, enable ratiometric imaging to eliminate motility artifacts, and permit

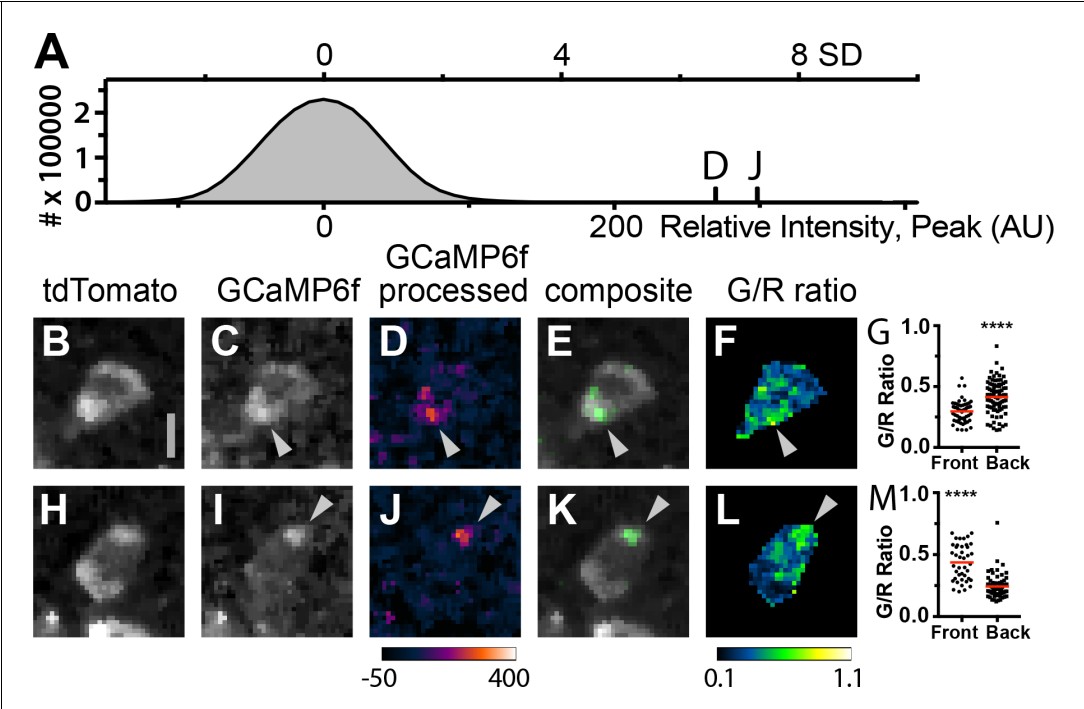

**Figure 10.** Subcellular Ca$^{2+}$ signals map to different regions of motile, adoptively transferred Cd4-Salsa6f (Hom) T cells. (A) Histogram of green channel pixel intensities from a representative region of a time-lapse image series of Cd4-Salsa6f (Hom) T cells in a wild-type lymph node after adoptive transfer. Vertical marks indicate the peak intensities of the fluorescence transients shown in D and J. (B–F, H–L) Two Cd4-Salsa6f (Hom) T cells imaged in a wild-type lymph node after adoptive transfer (same lymph node as in A). Red (B,H) and green (C,I) channel fluorescence images. (D,J) Corresponding pseudocolored green channel images processed as in *Figure 9—figure supplement 2*. (E,K) Corresponding composite image of gray pseudocolored red channel image with green channel processed image. (F,L) Ratiometric images of the green divided by the red channel fluorescence image. Gray arrowheads denote local Ca$^{2+}$ signals at the back (C–F) and front (I–L) of motile T cells. Look-up table for D and J) corresponds to Arbitrary Units; look-up table for F and L corresponds to green-to-red ratio. Both cells are oriented with their front toward the top of the image. Scale bar in A is 5 µm (applies to B-H, H–L). (G,M) Scatter plots of G/R ratio for individual pixels in the front and back of Cd4-Salsa6f T cells shown in B–F and H–L, respectively. Red lines indicate median values. (****) indicates p<0.0001, Mann Whitney test.

DOI: https://doi.org/10.7554/eLife.32417.023

convenient single-wavelength femtosecond excitation for two-photon microscopy. Salsa6f addresses a key weakness of single fluorescent protein-based GECIs by enabling tracking of motile cells and identification of cell morphology, even at basal Ca$^{2+}$ levels when GCaMP6f fluorescence is very weak. We generated a transgenic reporter mouse with Cre-dependent expression of Salsa6f, enabling Ca$^{2+}$ signals to be imaged in specific, genetically defined cell types. Transgenic expression of Salsa6f brings the power of ratiometric chemical Ca$^{2+}$ indicators to imaging cellular Ca$^{2+}$ signals amid the complex tissue environments found in vivo.

Salsa6f preserves the exceptional performance of GCaMP6f, which in the presence of high levels of Ca$^{2+}$ is as bright as the standard high-performing green fluorescent protein, EGFP (*Chen et al., 2013*). We find that Salsa6f possesses a dynamic range similar to GCaMP6f, and both are superior to FRET-based GECIs (*Heim et al., 2007*; *Thestrup et al., 2014*). The Ca$^{2+}$ affinity of Salsa6f, 160–300 nM, is well suited to detecting a variety of cellular Ca$^{2+}$ signals. Inclusion of tdTomato in Salsa6f enables ratiometric imaging, calibration, and measurement of Ca$^{2+}$ concentrations within cells. Moreover, Salsa6f is distributed uniformly throughout the cytosol; its exclusion from the nucleus provides reliable and selective reporting of cytosolic Ca$^{2+}$ signaling. This is in contrast to the recently developed PC::G5-tdT mouse strain in which the tdTomato is found throughout the cell but the separately expressed GCaMP5G is excluded from the nucleus (*Gee et al., 2014*).

We created a transgenic mouse strain in which Salsa6f is expressed under genetic control using the Rosa26$^{Cre}$ recombinase system, and we used this system to label immune cells that express Cd4. Salsa6f labeling enables readout of cytosolic Ca$^{2+}$ dynamics in T cells in vitro with high dynamic

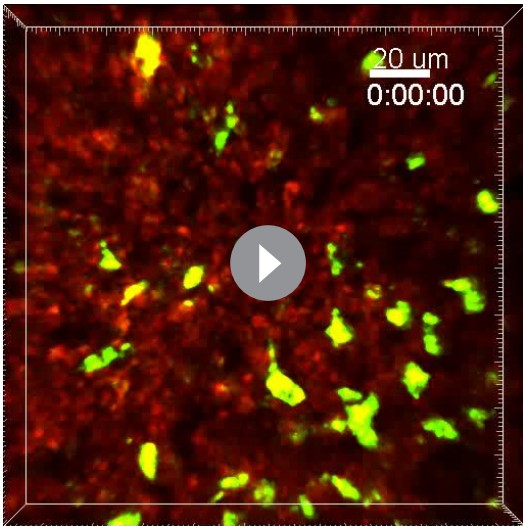

20 um

0:00:00

**Video 6.** Lymph nodes from Cd4-Salsa6f (Hom) mice exhibit cell-wide and subcellular Ca$^{2+}$ signals. Time shown in hr:min:s; images were acquired at 5 s intervals. Play back speed = 50 frames per second. Red channel is turned off after beginning to facilitate visualization of green signals. Video corresponds to *Figure 9B*.

DOI: https://doi.org/10.7554/eLife.32417.022

range, without the handling and potential toxicity associated with loading of chemical Ca$^{2+}$ indicators. Indeed, a concern with any Ca$^{2+}$ indicator is whether it may perturb cell function by buffering free Ca$^{2+}$ ions or other means. In Cd4$^+$ immune cells, we found no effects of Salsa6f expression, even in Cd4-Salsa6f (Hom) mice, with respect to cellular phenotype, cell proliferation, differentiation, and, in our companion paper (*Dong et al., 2017*), homing and T-cell motility.

Salsa6f was used to detect Ca$^{2+}$ influx due to direct activation of SOCE, TCR stimulation, and an activator of Piezo1 channels – the latter observed in T cells for the first time to our knowledge. We also detected differences in patterns of Ca$^{2+}$ signaling between naive T cells, Th17 cells, and iTregs. These experiments demonstrate the sensitivity, brightness, uniformity of labeling, and ease of detecting dynamic Ca$^{2+}$ signals using Salsa6f.

A primary advance of this work is to take the in vitro capabilities of an excellent Ca$^{2+}$ indicator and bring these into the realm of in vivo imaging. Within tissues, nearby cells exhibit differences, ranging from subtle to dramatic, in morphology, connectivity, and molecular profile. The red fluorescence of Salsa6f, combined with genetic Salsa6f labeling, associates these characteristics with readout of cellular Ca$^{2+}$ signaling. Both the tdTomato and GCaMP6f in Salsa6f are excited well by a 920 nm femtosecond pulsed wavelength for two-photon imaging, enabling visualization hundreds of micrometers deep into lymph nodes. The immune system poses additional challenges for imaging because the constituent cells are highly motile. Indeed, direct cell-to-cell interactions of motile immune cells form the basis of immune surveillance (*Mrass et al., 2010*; *Krummel et al., 2016*). We were able to readily identify red fluorescent Salsa6f T cells easily in intact lymph nodes following adoptive transfer. Our images reveal uniform red fluorescence labeling by Salsa6f with clear subcellular morphology in imaging sessions encompassing hundreds of time lapse images. Moreover, Salsa6f offers the opportunity not only to record fluctuations in relative Ca$^{2+}$ levels over time, but also to read out absolute Ca$^{2+}$ concentrations within cells. We have measured the affinity of Salsa6f in intact cells; in principle, use of this approach will allow other microscope systems to be calibrated for measuring absolute Ca$^{2+}$ concentrations with Salsa6f. Knowledge of absolute Ca$^{2+}$ concentrations is necessary to develop quantitative models of Ca$^{2+}$ signaling and cell behavior. Indeed, we demonstrate that clear Salsa6f ratio images can be generated from motile T cells in intact lymph nodes.

Our Salsa6f transgenic mouse line enables more sophisticated experimental approaches. One is the ability to detect rare Ca$^{2+}$ signaling events. The high brightness and dynamic range of modern GECIs like Salsa6f contribute to detection of rare Ca$^{2+}$ signaling events inside intact tissues or even whole transgenic animals (*Kubo et al., 2014*; *Portugues et al., 2014*). Ca$^{2+}$ signals have been previously recorded within lymph nodes in adoptively transferred T cells transgenically expressing doxycycline-inducible mCameleon (*Le Borgne et al., 2016*), GCaMP3 and dsRed as separate proteins (*Shulman et al., 2014*), and virally transduced YC-nano-50$^{CD}$ (*Liu et al., 2015*). Potential disadvantages of these probes include low dynamic range, nuclear expression of mCameleon and dsRed, the high Ca$^{2+}$ affinity of YC-nano-50$^{CD}$ (50 nM), and lack of ratiometric measurement. Imaging T cells transgenically expressing Salsa6f helps to overcome many of these limitations. Detecting rare events is made harder by inhomogeneities in cell populations of the lymph node as well as the movement of immune cells therein. Because of the one-to-one correspondence of tdTomato and GCaMP6f in Salsa6f, we were able to estimate and subtract resting GCaMP6f fluorescence even in motile cells.

This approach substantially improves the uniformity of the fluorescence background upon which rare Ca$^{2+}$ signaling events can be detected. Reliable and uniform cytosolic labeling contributes as well. Combined, these factors enabled us to detect not only sporadic cell-wide Ca$^{2+}$ elevations, but also sparkles, much smaller sporadic local Ca$^{2+}$ signals. The sensitivity and resolution of these images are sufficient to map local signals from intact lymph nodes to sub-regions of T cells. Moreover, while we focused upon the brightest local Ca$^{2+}$ signals to demonstrate their existence, we expect that Salsa6f will enable even lower intensity Ca$^{2+}$ signals to be linked to subcellular mechanisms and, ultimately, resulting cell behaviors. In a companion paper (*Dong et al., 2017*), we capitalize on the unique properties of Salsa6F to relate Ca$^{2+}$ signals, both global and local, to T-cell motility in the intact lymph node.

# Materials and methods

## Key resources table

| Reagent type (species) or resource | Designation | Source or reference | Identifiers | Additional information |
|---|---|---|---|---|
| Recombinant DNA reagent | Salsa6f | This paper | | Fusion of GCaMP6f to tdTomato via a V5 epitope linker (GCaMP6f-V5-tdTomato) |
| Recombinant DNA reagent | Gt(ROSA)26Sor5'-pCAG-FRT-LSL-Salsa6f-WPRE-bGHpA-AttB-FRT-NeoR-AttP-Gt(ROSA)26Sor3' cassette | This paper | | Salsa6f inserted into a Gt(ROSA)26Sor-pCAG-LSL-(Salsa6f)-WPRE-bGHpA-NeoR cassette. |
| Transgene (mouse) | *Gt(ROSA)26Sor*$^{pCAG-FRT-LSL-Salsa6f-WPRE-bGHpA-AttB-FRT-NeoR-AttP}$ | This paper | | Allele with the above cassette targeted to the ROSA26 locus. |
| Transgene (mouse) | *Gt(ROSA)26Sor*$^{pCAG-FRT-LSL-Salsa6f-WPRE-bGHpA-AttB/P}$ | This paper | | Same as above with Neomycin cassette deleted |
| Strain, strain background (mouse) | LSL-Salsa6f (F1), LSL-Sals6f (Hom) | This paper | | Salsa6f transgene targeted to Rosa26 locus in JM8.N4 mouse embryonic stem (ES) cells. Positive chimeras bred to R26ΦC31o mice to produce LSL-Salsa6f F1 founders and homozygotic LSL-Salsa6f (Hom) mice. See Materials and methods for details. |
| Strain, strain background (mouse) | Cd4-Salsa6f (Het), Cd4-Salsa6f (Hom) | This paper | | LSL-Salsa6f (Hom) mice crossed to *Cd4*$^{Cre}$ mice to produce heterozygotic and homozygotic *Salsa6f-Cd4*$^{Cre}$ mice (designated as Cd4-*Salsa6f* $^{±}$ and *Cd4-Salsa6f*$^{+/+}$ in the paper). |
| Strain, strain background (mouse) | *Cd4*$^{Cre}$ mice C57BL/6J | Jackson #017336 | | |
| Strain, strain background (mouse) | C57BL/6J | Jackson #000664 | | |
| Cell line (human) | HEK293A | Invitrogen (#R705-07) | | |
| Transfected construct (synthetic) | Salsa6f | This paper | | see above for *Salsa6f* gene |
| Transfected construct (synthetic) | G-GECO1, B-GECO1, GCaMP6f, GCaMP6m, GCaMP6s | Addgene | | |
| Antibody | anti-mouse IL-4, IL17A-APC (clone TC11-18H10.1), IFN-Pacific Blue (clone XMG1.2) | BioLegend | | |
| Antibody | Foxp3-PE (clone FJK16s) | ThermoFisher Scientific | | |
| Antibody | αCd3 and αCd28 | Invivogen | | |

*Continued on next page*

*Continued*

| Reagent type (species) or resource | Designation | Source or reference | Identifiers | Additional information |
|---|---|---|---|---|
| Antibody | αCd3 and αCd28 coated dynabeads | LifeTechnologies Corp. | | |
| Peptide, recombinant protein | recombinant human TGFβ1 | Tonbo Biosciences | | |
| peptide, recombinant protein | recombinant mouse IL-12, IL-23, IL-1β, TGFβ | BioLegend | | |
| Peptide, recombinant protein | recombinant human IL-2 | BioLegend | | |
| Commercial assay or kit | EasySep mouse naïve Cd4 T cell isolation kit | Stem Cell Technologies | | |
| Commercial assay or kit | EasySep mouse Cd4 T cell isolation kit | Stem Cell Technologies | | |
| Chemical compound, drug | Cell trace violet, eFluor 780 | ThermoFisher Scientific | | |
| Chemical compound, drug | Ionomycin, Thapsigargin, Retinoic Acid, PMA | Sigma Aldrich | | |
| Chemical compound, drug | Ghost dye 780 | Biolegend | | |
| Chemical compound, drug | Fura2-AM, Fluo-4 AM | ThermoFisher Scientific | | |
| Software, algorithm | ImageJ/Fiji | NIH | | |
| Software, algorithm | IMARIS | Bitplane | | |
| Other | 35 mm glass chamber | LabTek, ThermoFisher Scientific | | |
| Other | RPMI cell culture medium | Lonza | | |

## GECI screening and Salsa6f plasmid generation

Plasmids encoding GECIs (GECO and GCaMP6) were obtained from Addgene for screening in live cells. HEK 293A cells (Invitrogen- Life Technologies # R705-07) were screened for viruses and myco-plasma, split and frozen into working stocks, cultured using aseptic techniques, and used to evaluate candidate genetically encoded $Ca^{2+}$ indicators. Each probe was cotransfected with Orai1 and STIM1 into HEK 293A cells using Lipofectamine 2000 (Invitrogen, Carlsbad, CA) for 48 hr before screening on an epifluorescence microscope. For construction of Salsa6f, a plasmid for tdTomato (Addgene, Cambridge, MA) and the pEGP-N1 vector (Clontech, Mountain View, CA) was used as a backbone. GCaMP6f was amplified via PCR with N- and C-terminal primers (5' CACAACCGGTCGCCACCA TGGTCGACTCATCACGTC 3' and 5' AGTCGCGGCCGCTTTAAAGCTTCGCTGTCATCATTTGTAC 3') and ligated into pEGFP-N1 at the AgeI/NotI sites to replace the eGFP gene, while tdTomato was amplified via PCR with N- and C-terminal primers (5' ATCCGCTAGCGCTACCGGTCGCC 3' and 5' TAACGAGATCTGCTTGTACAGCTCGTCCATGCC 3') and ligated into the backbone at the NheI/ BglII sites. An oligo containing the V5 epitope tag was synthesized with sense and antisense strands (5' GATCTCGGGTAAGCCTATCCCTAACCCTCTCCTCGGTCTCGATTCTACG 3' and 5' GATCCG TAGAATCGAGACCGAGGAGAGGGTTAGGGATAGGCTTACCCGA 3') and ligated into the back-bone at the BglII/BamHI sites, linking tdTomato to GCaMP6f and creating Salsa6f. The amplified regions of the construct were verified by sequencing (Eton Bioscience Inc., San Diego, CA). This plas-mid, driven by the CMV promoter, was used for transient transfections in HEK 293A cells with Lipo-fectamine 2000 and in primary human T cells with Amaxa Nucleofection.

## Transgenic mouse generation and breeding

The transgenic cassette in *Figure 2B* was generated by inserting Salsa6f, from the plasmid described above, into the Ai38 vector (Addgene Plasmid #34883) and replacing GCaMP3. The final targeting vector included the CAG (cytomegalovirus early enhancer/chicken β-actin) promoter, an LSL sequence with LoxP-STOP-LoxP, the Salsa6f probe (tdTomato-V5-GCaMP6f), the woodchuck hepati-tis virus posttranscriptional regulatory element (WPRE), and a neomycin resistance gene (NeoR), all flanked by 5' and 3' Rosa26 homology arms of 1.1 and 4.3 kb. The targeting vector was linearized with PvuI and electroporated into JM8.N4 mouse embryonic stem (ES) cells of C57BL/6N back-ground. Following selection with G418, clones carrying the $Gt(ROSA)26Sor^{pCAG-FRT-LSL-Salsa6f-WPRE-bGHpA-AttB-FRT-NeoR-AttP}$ allele were screened by Southern blotting after digestion with HindIII for the

5' end or BglI for the 3' end. Four correctly targeted clones were expanded and checked by chromosome counting, then two clones with >90% euploidy were further expanded and injected into C57BL/6J blastocysts for implantation into pseudopregnant foster mothers. Presence of the Salsa6f transgenic cassette was detected in the resulting chimeric pups by PCR screening for the *Nnt* gene, as the initial JM8.N4 ES cells are $Nnt^{+/+}$ while the C57BL/6J blastocysts are $Nnt^{-/-}$. Finally, positive chimeras were bred to R26ΦC31o mice (JAX #007743) to remove the neomycin resistance gene flanked by AttB and AttP sites in the original transgenic cassette, and to produce F1 founders carrying the allele $Gt(ROSA)26Sor^{pCAG-FRT-LSL-Salsa6f-WPRE-bGHpA-AttB/P}$ at the Rosa26 locus. These F1 founders were then bred to homozygosity to generate LSL-Salsa6f (Hom) mice, and subsequently crossed to homozygotic $Cd4^{Cre}$ mice (JAX #017336) to generate Cd4-Salsa6f (Het) mice expressing Salsa6f only in T cells. *Cd4-Salsa6f* mice were further bred to generate homozygotic Cd4-Salsa6f (Hom) mice for increased Salsa6f expression and fluorescence.

## T-cell proliferation and differentiation

*For T-cell proliferation*: Cd4 T cells were isolated from spleen and lymph nodes of 6–10 week old mice using negative selection (StemCell Technologies, Cambridge, MA). CellTrace Violet (CTV)-labeled T cells were co-cultured with αCd3/Cd28 coated dynabeads (Life Technologies Corp., Grand Island, NY) at 1:1 ratio according to the manufacturer's protocol in a U bottom 96 well plate. *For T cell differentiation*: Naive Cd4 T cells were differentiated on activating polystyrene surface (Corning Inc., Corning, NY) with plate-bound αCd3 (2.5 µg/ml) and αCd28 (2.5 µg/ml) in the presence of cytokines for 6 days (*Yosef et al., 2013*). For Th1 differentiation: 25 ng/mL rmIL-12 (BioLegend, San Diego, CA), 10 µg/mL αmouse IL4 (Biolegend). For Th17 differentiation: 2.5 ng/mL rhTGF-β1 (Tonbo Biosciences, San Diego, CA), 50 ng/mL rmIL-6 (Tonbo Biosciences), 25 ng/ml rmIL-23 (BioLegend), and 25 ng/ml rmIL-β1 (BioLegend). For iTreg differentiation: 10 ng/mL rhTGF-β1, 100 units/mL of rmIL-2 (BioLegend), 5 µM Retinoic Acid (Sigma, St. Louis, MO).

## Flow cytometry

CTV dilution assay was performed in live cells (Fixable Viability Dye eFluor 780 negative gating; Thermofisher Scientific Inc., Grand Island, NY). To detect intracellular cytokines, 6 day differentiated cells were stimulated in with 25 ng of phorbol 12-myristate 13-acetate (PMA), 1 µg ionomycin (Sigma), and monensin (Golgistop BD biosciences) for 4 hr at 37°C. Dead cells were labeled with Ghost dye 780 (BioLegend), then washed, fixed, permeabilized using FoxP3 staining buffer set (Thermofisher Inc). The following antibodies were used to detect intracellular cytokines: IL-17A-APC (clone TC11-18H10.1, BioLegend); IFNγ-Pacific Blue (clone XMG1.2, BioLegend); Foxp3-PE (clone FJK16s, Thermofisher Scientific Inc.); in permeabilization buffer (eBioscience). Data were acquired using NovoCyte flow cytometer (ACEA Biosciences) and analyzed using FlowJo.

## T-cell preparation for live cell imaging

Cd4 T cells were activated by plating on six-well plates coated overnight with 2.5 µg/mL αCd3/α Cd28 (Invivogen, San Diego, CA) at 4°C. Cells were cultured in RPMI medium (Lonza) containing 10% FCS, L-glutamine, Non-essential amino acids, Sodium pyruvate, β-mercaptoethanol and 50 U/ mL of IL-2 at 37°C in 5% $CO_2$ incubator. Following 2 days of culture, cells were plated on either poly-L-lysine or 1 µg/mL α-Cd3/α28 coated 35-mm glass chambers (Lab-Tek, Thermofisher Inc.) for imaging. RPMI medium with 2% FCS and L-glutamine containing 2 mM $Ca^{2+}$ was used for imaging experiments. For experiments involving calibration and characterization of the Salsa6f probe in Cd4-Salsa6f cells, Ringer solution containing various concentrations of $Ca^{2+}$ was used. For $Ca^{2+}$ imaging of different T-cell subsets, Th17 cells and iTregs were differentiated as described above.

## Confocal imaging and analysis

For $Ca^{2+}$ imaging of $Cd4^+$ T cells from Cd4-Salsa6f mice, we used an Olympus Fluoview FV3000RS confocal laser scanning microscope, equipped with high-speed resonance scanner and the IX3-ZDC2 Z-drift compensator (Olympus Corp., Waltham, MA). Diode lasers (488 and 561 nm) were used for excitation, and two high-sensitivity cooled GaAsP PMTs were used for detection. Cells were imaged using the Olympus 40x silicone oil objective (NA 1.25), by taking five slice z-stacks at 2 µm/step, at 5 s intervals, for up to 20 min. Temperature, humidity, and $CO_2$ were maintained using a Tokai-Hit

WSKM-F1 stagetop incubator. Data were processed and analyzed using Imaris and ImageJ software. Calcium imaging experiments were done at 37°C on 2-day-activated $Cd4^+$ T cells from Cd4-Salsa6f (Het) mice, unless otherwise indicated. Salsa6f calibration experiments were done at room temperature.

## Two-photon microscopy

Lymph nodes images were acquired using a custom-built two photon microscope based on Olympus BX51 upright frame, Motorized ZDeck stage (Prior, Rockland, MA), with excitation generated by a tunable Chameleon femtosecond laser (Coherent, Santa Clara, CA) (*Miller et al., 2002*). The following wavelengths were used to excite single or combination of fluorophores: 920 nm to excite tdTomato and GCaMP6f; 1040 nm to excite tdTomato alone. 495 nm and 538 nm dichroic filters were arranged in series to separate blue, green and red signals. Two-photon excitation maxima of tdTomato and GCaMP6f are 1040 and 920 nm, respectively (*Drobizhev et al., 2011*; *Chen et al., 2013*). Using 1040 nm excitation, tdTomato signals were readily detected up to 300 μm depth; however, 1040 is not ideal to image Salsa6f because: 1) Collagen fibers generate second harmonic at 520 nm when excited with 1040 nm, which interferes with simultaneous detection of GCaMP6f (emission maxima, 509 nm) and 2) 1040 nm does not excite GCaMP6f (*Figure 9—figure supplement 1A*, top row). Alternatively, 920 nm optimally excites GCaMP6f, and excites tdTomato sufficiently, and Salsa6f signals were detected up to 300 μm depth, while second harmonic collagen signals (460 nm) can be easily separated into blue channel (*Figure 9—figure supplement 1A*, bottom row). Additionally, autofluorescent structures (LN resident DCs and fibroblastic reticular cells) show up as yellow bodies when excited with 920 nm, which serve as a guide to locate the T cell zone (*Figure 9—figure supplement 1B*). Therefore, 920 nm is the ideal two-photon excitation wavelength for simultaneous imaging of tdTomato and GCaMP6f as component parts of Salsa6f.

Lymph nodes were oriented with the hilum away from the water dipping microscope objective (Nikon 25x, NA 1.05). The node was maintained at 36–37°C by perfusion with medium (RPMI) bubbled with medical grade carbogen (95% $O_2$ and 5% $CO_2$) using a peristaltic heated perfusion system (Warner Instruments), with thermocouple-based temperature sensors placed next to the tissue in a custom built chamber. 3D image stacks of x = 250 μm, y = 250 μm, and z = 20 or 52 μm (4 μm step size) were sequentially acquired at 5 or 11 s intervals, respectively, using image acquisition software Slidebook (Intelligent Imaging Innovations) as described previously (*Matheu et al., 2015*). This volume collection was repeated for up to 40 min to create a 4D data set.

## Data analysis and statistical testing

Graphpad Prism was used for statistical analysis and generating figures. p values are indicated in figures: ns p>0.05, *p<0.05; **p<0.01; ***p<0.001; and ****p<0.0001.

## Detection of $Ca^{2+}$ signals in lymph nodes

Stacks of six optical sections 4 μm apart from the T-zone of Cd4-Salsa6f (Hom) lymph nodes were acquired once every 5 s at a resolution of 0.488 or 0.684 μm per pixel. Maximum intensity projections of 1 pixel radius median-filtered images were used for subsequent processing and analysis. Autofluorescent cells were identified by averaging the red or green time lapse image stacks and automated local thresholding (Bernsen five pixel radius) using the public domain image processing program ImageJ. Autofluorescent cell masks were dilated by four pixels, regions exhibiting less contrast and detail due to light scattering manually masked to produce the final time lapse image mask. Red (tdTomato) channel fluorescence from Salsa6f corresponding to green (GCaMP6f) channel resting state fluorescence was determined to be fivefold higher using our standard two-photon microscope acquisition settings. Final green images were produced by subtracting a 0.2x scaled red channel image, and subsequently subtracting out the average of all green channel time lapse images. The standard deviation (SD) of each masked green channel time lapse image stack was used to determine thresholds for local (sparkle) and cell-wide $Ca^{2+}$ events. Thresholds for detection of local and cell-wide $Ca^{2+}$ events were 5.4 and 2.1 SD and 1.4 $μm^2$ and 25 $μm^2$, respectively. Local $Ca^{2+}$ events were excluded from the set of sparkles if they coincided with a cell-wide event. Frequency of background events was calculated using a standard normal distribution with a Z-score corresponding to the average intensity of local events (6.5 SD), which was 1 in $2 \times 10^{10}$ pixels

(WolframAlpha). The front and back of motile T cells were traced manually from red channel TdTo-mato images. Individual pixel intensities from the front and back regions of interest were plotted and compared using the non-parametric Mann–Whitney test in Graphpad Prism.

## Acknowledgements

We acknowledge the UC Irvine Transgenic Mouse Facility for support in making the transgenic mouse, Dr. Grant MacGregor from the UC Irvine Department of Developmental and Cell Biology for excellent advice, and Dr. Jennifer Atwood of the Flow Core Facility supported by the UC Irvine Institute of Immunology. We also thank Andy Yeromin for advice on curve-fitting. This work was supported by an R21 grant AI117555, and RO1 grants NS14609 and AI121945, from the National Institutes of Health (MDC), and by a postdoctoral fellowship from the George E Hewitt Foundation for Medical Research (AJ).

## Additional information

### Funding

| Funder | Grant reference number | Author |
| --- | --- | --- |
| National Institutes of Health | AI117555 | Joseph L Dynes<br>Michael D Cahalan |
| National Institutes of Health | NS14609 | Michael D Cahalan |
| National Institutes of Health | AI121945 | Michael D Cahalan |

The funders had no role in study design, data collection and interpretation, or the decision to submit the work for publication.

### Author contributions

Tobias X Dong, Conceptualization, Data curation, Formal analysis, Supervision, Funding acquisition, Investigation, Methodology, Writing—original draft, Project administration, Writing—review and editing; Shivashankar Othy, Ian Parker, Resources, Data curation, Formal analysis, Investigation, Visualization, Methodology, Writing—original draft, Writing—review and editing; Amit Jairaman, Joseph L Dynes, Data curation, Formal analysis, Investigation, Visualization, Methodology, Writing—original draft, Writing—review and editing; Jonathan Skupsky, Conceptualization, Data curation, Formal analysis, Investigation, Methodology, Writing—original draft, Writing—review and editing; Angel Zavala, Resources, Data curation, Formal analysis, Investigation, Writing—review and editing; Michael D Cahalan, Conceptualization, Resources, Supervision, Funding acquisition, Writing—original draft, Project administration, Writing—review and editing

### Author ORCIDs

Tobias X Dong http://orcid.org/0000-0001-5500-7099
Shivashankar Othy http://orcid.org/0000-0001-6832-5547
Michael D Cahalan http://orcid.org/0000-0002-4987-2526

### Ethics

Human subjects: Human blood was prepared using support from the National Center for Research Resources and the National Center for Advancing Translational Sciences (NIH Grant UL1 TR000153). Animal experimentation: Use of blood samples from healthy human subjects has been approved by the University of California, Irvine Institutional Review Board (UCI IRB HS #1995-459). All animal procedures were approved by the UCI Institutional Animal Care and Use committee (IACUC) (protocol #1998-1366-11).

### Decision letter and Author response

Decision letter https://doi.org/10.7554/eLife.32417.026
Author response https://doi.org/10.7554/eLife.32417.027

## Additional files

**Supplementary files**
• Transparent reporting form
DOI: https://doi.org/10.7554/eLife.32417.024

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
