## [Decision Letter]

Thank you for submitting your article "T Cell Calcium Dynamics Visualized in a Ratiometric tdTomato-GCaMP6f Transgenic Reporter Mouse" for consideration by *eLife*. Your article has been reviewed by three peer reviewers, one of whom, Michael L Dustin (Reviewer #1), is a member of our Board of Reviewing Editors, and the evaluation has been overseen by Michel Nussenzweig as the Senior Editor. The following individuals involved in review of your submission have agreed to reveal their identity: Audrey Gerard (Reviewer #2); Ziv Shulman (Reviewer #3).

The reviewers have discussed the reviews with one another and the Reviewing Editor has drafted this decision to help you prepare a revised submission. Given that the work represents a technical development, we propose to consider your revised submission in the category of a Tools and Resources paper not as a Research Article. Although it would be labeled as such, it would appear with equal prominence along with all the other papers published at the same time.

Summary:

This is a methods paper describing Salsa6f mouse for quantitative tracking of Ca^2+^ in vitro and in vivo. This looks like a very useful mouse, particularly for adoptive transfer studies in the immune system, and when combined with different Cre's for different biological systems. The examples are quite detailed. For example, they examine Ca^2+^ signaling in T cell activation cultures on anti-CD3/anti-CD28 at 2 days of culture and how a "sustained oscillation" in a small fraction of cells, which would just not be possible with an organic dye in the time frame. They also show advantages over other efforts at ratiometric, genetically encoded fluorescent proteins. They don't show comparison to the best recent FRET reporters, but it’s pretty clear they will not be able to compare in terms of the dynamic range of the green signal. The most biologically novel finding is related to demonstrating an effect of Piezo activator Yoda on Calcium responses in T cells in vitro. While this probably could have been done with Fura-2, this fits with the investigators interested in motility, and opens the possibility of examining this in vivo. This data is suggestive because it’s just one type of experiment with a small molecule activator in a cell type where the activator has not been validated to target only Piezo channels. Finally, they reveal subcellular Ca^2+^ spikes in the rear of migrating T cell in intact lymph nodes imaged ex vivo, which has not been reported before- so a new phenomenon that further demonstrates the power of the system. There is some concern that the sub cellular localisation of this signal may reflect the distribution of cytoplasm for the authors should be more conservative about interpreting this unless they want to provide clear quantification showing that the green/red ratio shows sub cellular Ca^2+^ over what is expected based on signal to noise. One can immediately begin to formulate hyptotheses about what this could be and the model will certainly be useful in determining the basis of these events.

Essential revisions:

The authors should provide quantification of the data described in relation to Figure 10 breakdown of the different patterns with number mice/LN analyzed and number of events observed. If statistically valid evidence of sub cellular Calcium is found then this should be discussed in the context of other invert studies that look at such local Ca^2+^ signals. If it turns out that this is not possible, a more conservative interpretation should be presented and the advantages of the reporter would still be clear.

---

## [Author Response]

Essential revisions:The authors should provide quantification of the data described in relation to Figure 10 breakdown of the different patterns with number mice/LN analyzed and number of events observed. If statistically valid evidence of sub cellular Calcium is found then this should be discussed in the context of other invert studies that look at such local Ca^2+^ signals. If it turns out that this is not possible, a more conservative interpretation should be presented and the advantages of the reporter would still be clear.

We believe that a detailed characterization of patterns of cellular behavior in relation to sparkles would be beyond the scope of this paper. Instead, we focused on ensuring that the signal-to-noise issues are addressed thoroughly and that the green/red ratio quantitation validates that sparkles are truly subcellular local signaling events. We are not certain about which “invert” studies the comment refers to. Local subcellular Ca^2+^ signals have been detected using chemical indicators in a variety of cell types over the last 20-30 years. We further note that the accepted companion paper illustrates the relation of sparkles to cellular motility (Figure 7), presents an independent method for analyzing sparkles (Figure 8), and a peptide-MHC-dependent mechanism (Figure 9). Figure 9 of the current manuscript illustrates detection of abundant sparkles in homozygous Cd4-Salsa6f lymph node; image processing minimized fluctuations in background fluorescence in order to identify cell-wide Ca^2+^ signals and sparkles.

In response to the reviewer comment, Figure 10 was revised and now includes a pixel intensity histogram, and two illustrations of how sparkles are localized in adoptively transferred Cd4-Salsa6f^+/+^ T cells. The amplitude histogram (new Figure 10) illustrates sparkle signals that are more than 5 standard deviations above background noise. The remaining panels illustrate image processing on two representative sparkles to show that green/red ratios exhibit regional variation that are statistically significant beyond p values of 0.0001. The text in the Materials and methods has been amended to clarify sparkle identification, and we apologize for the confusion. As motile T cells extend, they adopt a polarized morphology, in which most cytoplasm is concentrated at the front and back with little on the sides of the cell (see our accepted companion *eLife* motility manuscript for additional example images). Because of this arrangement of cytoplasm, it is possible that apparent sparkles might arise from global Ca^2+^ signals in which asymmetry in the distribution of T cell cytoplasm limits detection of fluorescence from other regions of the cell, or alternatively from polarized T cells in which half of the cell lies outside the imaging field. Bright local Ca^2+^ events (>5.4 SD) were excluded from the set of sparkles if they coincided with a cell-wide event, even if the intensity of cell-wide event was much lower (2.1 SD threshold). We show that sparkles are not subregions of cells with slightly higher amounts of fluorescence that we produced by inappropriately thresholding cell-wide Ca^2+^ signals. To address this potential concern, we selected example cells in which the front and back of the cells were clearly visible. We used a simple quantitative approach to establish that Ca^2+^ signals detected can selectively occupy either the front or back of these cells. Local signals with peak SNR of 6.6 and 7.3 were chosen, comparable in size to local signals reported in whole Cd4-Salsa6f lymph nodes (Figure 9). For both cells in Figure 10, five views are now shown: the median filtered maximum projection red TdTomato and green GCaMP6f channels (new Figure 10); the green GCaMP6f channel processed image (new Figure 10); two-channel composite images which have been green channel thresholded but are a clear way to visualize the locations of Ca^2+^ signals in cells (Figure 10; compare to panels D and J to ensure that the distribution of green fluorescence has not compromised by being displayed in this manner); and the G/R ratio (Figure 10). G/R ratio largely reflects the location of detected sparkles.